# Nash: Neural Adaptive Shrinkage for Structured High-Dimensional Regression

## Abstract

Sparse linear regression is a fundamental tool in data analysis. However, traditional approaches often fall short when covariates exhibit structure or arise from heterogeneous sources. In biomedical applications, covariates may stem from distinct modalities or be structured according to an underlying graph. We introduce *Neural Adaptive Shrinkage* (Nash), a unified framework that integrates covariate-specific side information into sparse regression via neural networks. Nash adaptively modulates penalties on a per-covariate basis, learning to tailor regularization without cross-validation. We develop a variational inference algorithm for efficient training and establish connections to empirical Bayes regression. Experiments on real data demonstrate that Nash can improve accuracy and adaptability over existing methods.

## 1 Introduction

Regularization techniques for linear models have been central in data analysis for decades (Hoerl & Kennard, 1970; Tibshirani, 1996; Zou & Hastie, 2005). They remain central in modern data analysis as they are competitive approaches when the sample size is limited and the covariates are high-dimensional (Horvath & Raj, 2018; Bohlin et al., 2016; Horvath & Raj, 2018; Haftorn et al., 2021). Despite their popularity, these methods often fall short when dealing with heterogeneous covariates that exhibit structural properties, such as nominal, ordinal, spatial, or graphical data. Classical regularization methods like Lasso (Tibshirani, 1996) typically apply uniform penalties across all covariates, which can be suboptimal when diverse predictor types are present in the covariate matrix (e.g, different genetic modalities). Real-world problems often benefit from tailored regularization that leverages the covariate side information, such as geographical proximity (Devriendt et al., 2021) or type biological measurements (Boulesteix et al., 2017). On the other hand, the existing methods that leverage covariate side information (Tibshirani et al., 2005; Yuan & Lin, 2006; Yu et al., 2016; Boulesteix et al., 2017) are often limited by their application-specific nature and reliance on cumbersome cross-validation for hyperparameter selection (Tibshirani et al., 2005; Yuan & Lin, 2006).

In this work, we introduce *neural adaptive shrinkage* (Nash), a novel regression model framework that can leverage neural networks to automatically learn the form of the penalty and select the amount of regularization without using cross-validation or approximate methods. Hence, alleviating the limitations listed above. We fit Nash using a novel variational inference empirical Bayes (VEB) method called split VEB, originally introduced for smoothing over-dispersed Poisson counts (Xie, 2023), that we adapt here for high-dimensional Gaussian linear models. When no side information is available, our approach corresponds to optimizing the lower bound of a recently proposed model by Kim et al. (2024) and has similar computation complexity $O((n + K)p)$. However, our learning algorithm is much simpler than the one proposed by Kim et al. (2024) and allows easy integration of machine learning approaches for penalty learning (e.g., neural net, xgboost Chen & Guestrin (2016)). Hence, Nash is both an extremely efficient high-dimensional regression method when no side information is present and a very flexible alternative when side information is available. We demonstrate that Nash is a highly competitive framework through a comprehensive study on real data examples.

## 2 PREVIOUS WORKS AND CONTRIBUTION

**Previous works** have mostly focused on two main types of side information on the covariate. The first type corresponds to groups (e.g. DNA methylation vs genotype data (Boulesteix et al., 2017)) or hierarchical information on the covariate; these works include group Lasso (Yuan & Lin, 2006) and other of its variations (Gertheiss & Tutz, 2010; Tutz & Oelker, 2017; Oelker & Tutz, 2017) and the IPF Lasso (Boulesteix et al., 2017). Essentially, these methods extend classical regularized techniques for linear models by using different additive sub-penalties that depend on the group/hierarchy of the covariates. The second type of covariate side information leveraged in penalized regression is graphs (Tibshirani et al., 2005; Tibshirani & Taylor, 2011). Spanning from simple L0 graph filtering problem such as Fused Lasso (Tibshirani et al., 2005) to more complex graphical structure that can be handled by the GEN Lasso (Tibshirani & Taylor, 2011) and more recent variations (Yu et al., 2016; Devriendt et al., 2021) that can fit a mix of the different penalties above within a single framework.

**Our contribution.** While combining neural networks with linear regression is not new (Okoh et al., 2018; Nalisnick et al., 2019; Lemhadri et al., 2021), existing methods focus on hybrid models (Okoh et al., 2018; Nalisnick et al., 2019) or learning link functions (Lemhadri et al., 2021) rather than learn the penalty itself. Our work differs substantially from the previous works listed above. To our knowledge, this is the first work to propose the use of a neural network to incorporate covariate side information when learning the penalty function in linear regression. Our work is much more assumption-lean compared to previous works, as Nash can leverage any side information that is processable by a neural net. Additionally, we propose a novel low-complexity variational approximation for empirical Bayes in multiple linear regression. The resulting learning algorithm is a simple and effective iterative procedure, akin to ADMM or proximal algorithms (Polson et al., 2015).

## 3 PROBLEM DEFINITION

### 3.1 VARIATIONAL EMPIRICAL BAYES FOR THE NASH MODEL

The Nash model is defined as follows:

$$\boldsymbol{y}|\boldsymbol{X}, \boldsymbol{\beta}, \sigma^2 \sim N(\boldsymbol{X}\boldsymbol{\beta}, \sigma^2) \tag{1}$$
$$\beta_j \sim N(b_j, \sigma_0^2) \tag{2}$$
$$b_j \sim g(\boldsymbol{d}_j, \boldsymbol{\theta}) \tag{3}$$

where $\boldsymbol{y}$ is a response vector of length $n$, $\boldsymbol{X}$ is an $n \times p$ matrix, where $p$ can be much larger than $n$ (i.e., $p \gg n$), and $\boldsymbol{x}_j$ is the $j^{th}$ column of $\boldsymbol{X}$. The terms $\sigma^2 > 0$ and $\sigma_0^2 > 0$ are strictly positive variance parameters. The vector $\boldsymbol{d}_j$ corresponds to side information on column $j$. The function $g(\cdot, \cdot)$ belongs to a certain class of functions $\mathcal{G}$ and takes $\boldsymbol{d}_j$ (side information) as its first argument and $\boldsymbol{\theta}$ (parameters) as its second argument. For any tuple $(\boldsymbol{d}_j, \boldsymbol{\theta})$, $g(\boldsymbol{d}_j, \boldsymbol{\theta})$ defines a distribution with a density, denoted as $g(b_j; \boldsymbol{d}_j, \boldsymbol{\theta})$ at the point $b_j$.

For simplicity, we assume that $\boldsymbol{y}$ is scaled, centered with unit variance, and similarly that each columns of $\boldsymbol{X}$ (i.e., $\|\boldsymbol{x}_j\| = 1$ and $\mathbb{E}(x_j) = 0$ for all $j = 1, \ldots, p$). Note that we do not model the intercept in equation 1, as centering $\boldsymbol{y}$ and $\boldsymbol{X}$ prior to model fitting accounts for it, and it is straightforward to recover the effect for the unscaled $\boldsymbol{X}$ (Chipman et al., 2001).

We assume that for each predictor $\boldsymbol{x}_j$ in $\boldsymbol{X}$, we observe some side information $\boldsymbol{d}_j$. We intentionally remain vague on the form of the side information $\boldsymbol{d}_j$, with the only constraint being that $\boldsymbol{d}_1, \ldots, \boldsymbol{d}_p$ can be processed by a neural network (e.g., images, tokens, graph matrices). For ease of presentation, we assume that we can store $\boldsymbol{d}_1, \ldots, \boldsymbol{d}_p$ in a matrix $\boldsymbol{D}$ of size $p \times k$. The case without any side information can be recovered by setting $\boldsymbol{d}_1 = \boldsymbol{d}_2 = \ldots = \boldsymbol{d}_p$, i.e., constant side information.

Assuming that $\sigma^2$ and $\sigma_0^2$ are known, solving equation 1 in an Empirical Bayes (EB) fashion involves the following steps:

1. Learning the parameter $\boldsymbol{\theta}$ of the function $g(\cdot, \cdot) \in \mathcal{G}$ via maximum marginal likelihood $\mathcal{L}(\boldsymbol{\theta})$

$$\hat{\boldsymbol{\theta}} = \arg \max_{\boldsymbol{\theta}} \mathcal{L}(\boldsymbol{\theta}) \tag{4}$$

$$= \arg \max_{\boldsymbol{\theta}} \int p(\boldsymbol{y}|\boldsymbol{X}, \boldsymbol{\beta}, \sigma^2) \prod_j p(\beta_j|b_j, \sigma_0^2) g(b_j; \boldsymbol{d}_j, \theta) \, db_j \tag{5}$$

2. Compute the posterior distribution

$$p_{\text{post}}(\boldsymbol{\beta}, \boldsymbol{b}) = p(\boldsymbol{\beta}, \boldsymbol{b}|\boldsymbol{y}, \boldsymbol{X}, \boldsymbol{D}, \sigma^2) \propto p(\boldsymbol{y}|\boldsymbol{X}, \boldsymbol{\beta}, \sigma^2) \prod_j p(\beta_j|b_j, \sigma_0^2) g(b_j; \boldsymbol{d}_j, \hat{\boldsymbol{\theta}}) \tag{6}$$

Wang & Stephens (2021) and Kim et al. (2024) study similar problems in the case where $g$ does not depend on side information $\boldsymbol{d}$ (i.e., $g(\boldsymbol{d}_j, \boldsymbol{\theta}) = g(\boldsymbol{\theta})$ for all $j$). They note that even in this case, both steps described above are computationally intractable except in some very special cases. This problem becomes even more challenging when we allow the prior $g$ to depend on side information.

## 3.2 SPLIT VARIATIONAL INFERENCE

Given that we aim to fit model equation 1 in a tractable and efficient way, we propose fitting equation 1 via split VEB (Xie, 2023) using a candidate posterior of the form:

$$q(\boldsymbol{\beta}, \boldsymbol{b}) = \prod_j^P q_{\beta_j}(\beta_j) q_{b_j}(b_j) \tag{7}$$

The main idea behind split VEB is to decouple the prior/penalty learning step (step 1) from the posterior computation step (step 2). Our primary quantity of interest is the posterior of $\boldsymbol{b}$. However, using $\boldsymbol{b}$ directly in the linear predictor results in a coupled prior/posterior update as in Kim et al. (2024). To alleviate this problem, we essentially introduce a latent variable $\boldsymbol{\beta}$ that allows splitting the ELBO into two parts that are separately updated (see equation 8). At a high level, split VEB allows deriving a coordinate ascent that essentially iterates between solving two simple problems similar to optimization techniques (e.g., ADMM or proximal algorithms Polson et al. (2015)).

**Form of the ELBO** Using the candidate posteriors of the form 7 leads to an ELBO of the following form for the Nash model

$$F(q_{\boldsymbol{\beta}}, q_{\boldsymbol{b}}, g, \sigma^2, \sigma_0^2)_{Nash} = \sum_i \mathbb{E}_{q(\boldsymbol{\beta}, \boldsymbol{b})} \left[ \log \frac{p(y_i|\boldsymbol{x}_i, \boldsymbol{\beta}, \sigma^2)}{q_{\boldsymbol{\beta}}(\boldsymbol{\beta})} \right] + \sum_j \mathbb{E}_{q(\boldsymbol{\beta}, \boldsymbol{b})} \left[ \log p(\beta_j|b_j, \sigma_0^2) \right] + \tag{8}$$

$$\sum_j \mathbb{E}_{q(\boldsymbol{\beta}, \boldsymbol{b})} \left[ \log \frac{g(b_j; \boldsymbol{d}_j, \boldsymbol{\theta})}{q_{b_j}(b_j)} \right] \tag{9}$$

Where $q(\boldsymbol{\beta}, \boldsymbol{b})$ is the mean-field variational distribution as defined in 7.

**High-Level Coordinate Ascent Update for Nash** Let $\bar{\beta}_j = \mathbb{E}_q(\beta_j)$ denote the expected value of $\beta_j$ with respect to $q$, and $\bar{b}_j = \mathbb{E}_q(b_j)$ denote the expected value of $b_j$. We define $\bar{\boldsymbol{r}} = \boldsymbol{y} - \boldsymbol{X}\bar{\boldsymbol{\beta}}$ as the vector of expected residuals with respect to $q$. Let $\boldsymbol{X}_{-j}$ be the design matrix excluding the $j^{th}$ column, and $q_{-j}$ denote all factors $q_{j'}$ except factor $j$. The expected residuals accounting for the linear effect of all variables other than $j$ are given by:

$$\bar{\boldsymbol{r}}_j = \boldsymbol{y} - \boldsymbol{X}_{-j}\bar{\boldsymbol{\beta}}_{-j} = \boldsymbol{y} - \sum_{j' \neq j} \boldsymbol{x}_{j'}\bar{\beta}_{j'} \tag{10}$$

1. **Update for $q^*_{\beta_j}$:** the coordinate ascent update $q^*_{\beta_j} = \arg\max_{q_{\beta_j}} F_{\text{Nash}}(g, q, \sigma^2, g, q, \sigma_0^2)$ is obtained by computing the posterior using

$$p(\bar{r}_j|x_j, \beta_j, \sigma)p(\beta_j|\bar{b}_j, \sigma_0^2)$$

   This is a simple posterior computation due to conjugacy and has a closed form that only requires computing the ordinary least square (OLS) regression of $x_j$ on $\bar{r}_j$ (see section 3.2.1).

2. **Update for $(g^*, q^*_b)$:** The coordinate ascent update

$$(g^*, q^*_b) = \arg\max_{g,q_b} F(q_{\boldsymbol{\beta}}, q_{\boldsymbol{b}}, g; \sigma^2)_{\text{Nash}}$$

   is obtained by fitting a neural net with the following objective function:

$$\hat{\boldsymbol{\theta}} = \arg\max_{\boldsymbol{\theta}} \mathcal{L}(\boldsymbol{\theta}) \tag{11}$$

$$= \arg\max_{\boldsymbol{\theta}} \prod_{j=1}^{p} \int \mathcal{N}(\bar{\beta}_j; b_j, \sigma_0^2)\, g(b_j; \boldsymbol{d}_j, \boldsymbol{\theta})\, db_j \tag{12}$$

   Then, by computing the posterior of

$$p(b_j|\bar{\beta}_j, \boldsymbol{d}_i, \sigma_0^2) \propto \mathcal{N}(\bar{\beta}_j; b_j, \sigma_0^2)\, g(b_i; \boldsymbol{d}_i, \hat{\boldsymbol{\theta}})$$

   for each $b_j$, which is also a simple posterior computation.

3. **Update for $\sigma^2, \sigma_0^2$**
   (a) $(\sigma^2)^* = \arg\max_{\sigma^2} F(q_{\boldsymbol{\beta}}, q_{\boldsymbol{b}}, g, \sigma^2, \sigma_0^2)_{\text{Nash}}$
   (b) $(\sigma_0^2)^* = \arg\max_{\sigma_0^2} F(q_{\boldsymbol{\beta}}, q_{\boldsymbol{b}}, g, \sigma^2, \sigma_0^2)_{\text{Nash}}$

The first step is a direct consequence of the work by Kim et al. (2024) (see Appendix for more details), the second step results from our splitting approach, and the last step is a standard coordinate ascent variational inference (CAVI) step. We provide the closed-form formulas for both $\sigma^2$ and $\sigma_0^2$ in the supplementary section A.1, and we describe the overall learning process in the Appendix, Algorithm 1.

**Choice of $\mathcal{G}$**  For clarity, suppose that $g(\cdot, \cdot)$ belong to a family of distributions $\mathcal{G}$ that have the following form:

$$g(\boldsymbol{d}_j, \boldsymbol{\theta}) = \sum_{m=0}^{M} \pi_m(\boldsymbol{d}_j, \boldsymbol{\theta}) g_m \tag{13}$$

$$\boldsymbol{\pi}(\boldsymbol{d}_j, \boldsymbol{\theta}) = (\pi_0(\boldsymbol{d}_j, \boldsymbol{\theta}), \dots, \pi_M(\boldsymbol{d}_j, \boldsymbol{\theta})) \tag{14}$$

where $g_m$ are fixed known distributions (e.g., $g_0 = \delta_0$ and $g_m = \mathcal{N}(0, \sigma_m^2)$ with $\sigma_m^2 < \sigma_{m+1}^2$ for all $m > 0$). Then $\boldsymbol{\pi}(., \boldsymbol{\theta})$ is a neural network that takes side information $\boldsymbol{d}_j$ as input and outputs a vector of probabilities $(\pi_0(\boldsymbol{d}_j, \boldsymbol{\theta}), \dots, \pi_M(\boldsymbol{d}_j, \boldsymbol{\theta}))$ that sum to 1 (e.g., using a softmax function). Under this model, the loss for $\boldsymbol{\theta}$ has the following simple form:

$$\hat{\boldsymbol{\theta}} = \arg\max_{\boldsymbol{\theta}} \sum_{j=1}^{P} \log \sum_{m=0}^{M} \pi_m(\boldsymbol{d}_j, \boldsymbol{\theta}) L_{jm} \tag{15}$$

where $L_{jm}$ is defined as:

$$L_{jm} = \int p(\bar{\beta}_j|b_j) g_m(b_j)\, db_j \tag{16}$$

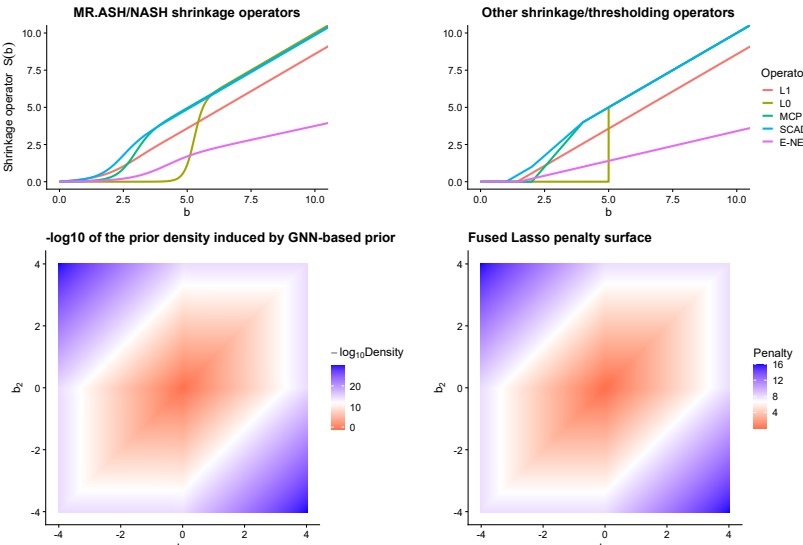

Figure 1: Upper panel: Adaptation of Figure 1 from Kim et al. (2024), showcasing that posterior mean shrinkage operators (left panel) for different choices of $\sigma_1^2, \ldots, \sigma_M^2$ and $\pi_0, \ldots, \pi_M$ can mimic the shrinkage operators from some commonly used penalties (right-hand panel). Bottom panel left: Illustration of how Nash can mimic fused Lasso penalty when used with a graph neural net prior-based. The left image presents the induced prior density from equation 24, allowing Nash to mimic the fused Lasso penalty ( using $s_1 = 0.45$ and $s_2 = 0.15$). Bottom right panel, penalty surface of the fused Lasso (i.e., $|b_1 + b_2| + |b_1 - b_2|$).

This represents the marginal likelihood of $\bar{\bar{\beta}}_j$ under mixture component $m$. For Gaussian mixture components $g_m = \mathcal{N}(0, \sigma_m^2)$, we have:

$$L_{jm} = \mathcal{N}(\hat{\beta}_j; 0, \sigma_0^2 + \sigma_m^2) \tag{17}$$

These integrals often cannot be computed analytically for other priors and error models. However, $(L_{jm})$ are simple one-dimensional integrals that are fast to approximate. It is straightforward to extend this model to use more complex distribution mixtures such as Mixture Density Networks Bishop (1994) or Graph Mixture Density Networks Errica et al. (2021) (see section 4 for more details). More generally, $g(\cdot, \cdot)$ can be any probabilistic model (e.g., Gaussian Process, but potentially more complex models) for which the loss in 12 can be evaluated and the posterior $p(b_j|\bar{\beta}_j, \boldsymbol{d}_i, \sigma_0^2) \propto \mathcal{N}(\bar{\beta}_j; b_j, \sigma_0^2) g(b_i; \boldsymbol{d}_i, \hat{\boldsymbol{\theta}})$ can be computed. For computational efficiency, it is useful that both of these steps can be evaluated via closed-form formulas. Note that when $g$ does not depend on the covariate, then step 2) in the coordinate ascent described above corresponds to an empirical Bayes normal mean problem (EBNM; see Willwerscheid et al. (2024) for an overview, and (Robbins, 1956; Efron, 2019; Stephens, 2017) for classical statistical papers on this topic). In this case, fitting $g$ as in 13 with fixed $g_m(\cdot)$ distributions corresponds to estimating the mixture proportions for the different mixture components $g_m(\cdot)$. Using fixed distributions is particularly practical as it allows efficient estimation of the mixture components $(\pi_0, \ldots, \pi_M)$ via sequential quadratic programming, which is often achieved in sub-linear time (in terms of $p$), see Kim et al. (2019).

### 3.2.1 ON THE UPDATE FOR $\beta$

A careful reader will notice that the update for $\boldsymbol{\beta}$ can actually be solved exactly, without using an approximate posterior as in 7. Split VEB leads to an update for $\boldsymbol{\beta}$ that corresponds to a Bayesian ridge regression $\boldsymbol{y}|\boldsymbol{X}, \boldsymbol{\beta}, \sigma^2 \sim N(\boldsymbol{X}\boldsymbol{\beta}, \sigma^2)$ with a prior on $\boldsymbol{\beta} \sim N(\boldsymbol{b}, \sigma_0^2 I_p)$. The posterior of $\boldsymbol{\beta}$ has a well-known closed form (Hoerl & Kennard, 1970). However, computing this posterior requires inverting a matrix, resulting in $O(np^2 + p^3)$ operations to compute the exact posterior (or $O(n^2p + n^3)$ operations using the dual form, via Woodbury formula (Saunders et al., 1998)). Because of conjugacy

and the assumption that the columns of $\boldsymbol{X}$ are centered and scaled, the update at iteration $t+1$ for $\boldsymbol{\beta}_j$ under 7 is given by $\boldsymbol{\beta}_j^{t+1} = \omega \boldsymbol{x}_j^\intercal \bar{\boldsymbol{r}}_j^{t+1} + (1 - \omega)\bar{b}_j^t$. Here, $\bar{r}_j^{t+1}$ is the expected residual at iteration $t+1$ as defined in 10, $\bar{b}_j^t$ is the posterior mean of $b_j$ under 7 at iteration $t$, and $\boldsymbol{x}_j^\intercal \bar{\boldsymbol{r}}_j$ corresponds to the maximum likelihood estimate (MLE) of the effect of $\boldsymbol{x}_j$ on $\bar{\boldsymbol{r}}_j^{t+1}$ due to scaling. The term $\omega$ is defined as $\omega = \frac{(n-1)\sigma_0^2}{\sigma^2 + (n-1)\sigma_0^2}$ as due to scaling $\boldsymbol{x}_j^\intercal \boldsymbol{x}_j = n - 1$ . Therefore, each update for a $\beta_j$ corresponds to a scalar product between two vectors, resulting in a coordinate ascent algorithm that has a complexity of $O(np)$, which is significantly smaller than $O(n^2p + n^3)$ or $O(np^2 + p^3)$.

### 3.2.2 AN AUTOREGRESSIVE UPDATE FOR $g$ WITH AUTO-ADAPTIVE DAMPENING

Given that in practice both $\sigma^2$ and $\sigma_0^2$ are being updated (see steps 3a and 3b in the coordinate ascent algorithm above), the resulting updates for $g$ and $q_b$ correspond to fitting a series of autoregressive covariate-moderated empirical Bayes normal mean problems (cEBNM) that have the following form:

$$\omega_t \hat{\beta}_{j_{MLE}}^{t+1} + (1 - \omega_t)\bar{b}_j^t \sim N(b_j^{t+1}, \sigma_{0,t}^2), \tag{18}$$

$$b_j^{t+1} \sim g(\boldsymbol{d}_j, \boldsymbol{\theta}^{t+1}). \tag{19}$$

Here, $\omega_t = \frac{(n-1)\sigma_{0t}^2}{\sigma_t^2 + (n-1)\sigma_{0t}^2}$, where $\sigma_{0t}$ is the value of $\sigma_0^2$ at iteration $t$ (the same goes for $\sigma_t^2$). Equation 18 arises from basic Bayesian computation, yet it leads to an update that is simple to interpret. At each update for $g$ and $q_b$, the model uses a proportion $\omega_t$ of novel evidence while retaining $1 - \omega_t$ of the previous update. Given that $\sigma_t^2$ and $\sigma_{0t}^2$ are maximized by Nash's ELBO, the parameter $\omega_t$ can be viewed as a data-driven dampening parameter for learning $g$ and $q_b$.

### 3.3 COMPARISON AND CONNECTION WITH MR.ASH

Our work is closely related to the multiple regression with adaptive shrinkage (mr.ash) proposed by Kim et al. (2024), but it differs in two key aspects. The most notable difference is that mr.ash cannot handle side information. A more technical yet important difference is our learning algorithm. Kim et al. (2024) use a standard CAVI for fitting VEB approximation of mr.ash, which results in a coordinate ascent algorithm that requires updating the prior $g$ whenever updating $q_{b_j}$. Thus mr. ash's update for $g$ corresponds to an M-step (Dempster et al., 1977). Because the posterior and the prior learning steps are not decoupled in the mr.ash variational formulation, the resulting CAVI requires updating the prior $g$, $p$ times per CAVI update. Split VEB allows decoupling these two problems, leading to a coordinate ascent for Nash that is notably more efficient, as it only requires updating the prior $g$ once per coordinate ascent update. While this nuance may appear subtle at first, it turns out to be crucial when side information is present. Using split VEB allows fitting Nash with a single update of the neural net parameters $\theta\ (g(\cdot, \theta))$ per coordinate ascent update iteration. In contrast, adapting mr.ash would require updating the neural net $p$ times per CAVI update, which is not practical when $p$ is large.

$$
\begin{array}{cc}
\text{mr.ash} & \text{Nash, without side information} \tag{20}
\end{array}
$$

$$
\boldsymbol{y}|\boldsymbol{X}, \boldsymbol{b}, \sigma^2 \sim N(\boldsymbol{Xb}, \sigma^2) \qquad \boldsymbol{y}|\boldsymbol{X}, \boldsymbol{\beta}, \sigma^2 \sim N(\boldsymbol{X\beta}, \sigma^2) \tag{21}
$$

$$
b_j \sim g \qquad \beta_j \sim N(b_j, \sigma_0^2) \tag{22}
$$

$$
b_j \sim g \tag{23}
$$

These two works are related, as fitting Nash with split VEB when no side information is provided corresponds to optimizing a lower bound of mr.ash's Evidence Lower Bound (ELBO), when using $g \in \mathcal{G}$ from the same family of distributions when fitting both models (see section A.2 for a formal proof. We also provide in supplementary material **Algorithm 2** a high-level description of the key differences between Nash and mr.ash fitting procedures.

# 4 Connection to Penalized Linear Regression and Beyond

Kim et al. (2024) showed that mr.ash (and therefore Nash) can be viewed as a penalized linear regression (PLR) problem. When using an adaptive shrinkage prior (ash, (Stephens, 2017)) of the form $g = \pi_0 \delta_0 + \sum_{m=1}^{M} \pi_m N(0, \sigma_m^2)$, different choices of $(\pi_0, \ldots, \pi_M)$ corresponding to different penalties such as Ridge regression (Hoerl & Kennard, 1970), L0Learn (Hazimeh et al., 2023), Lasso (Tibshirani, 1996), Elastic Net (Zou & Hastie, 2005), the smoothly clipped absolute deviation (SCAD) penalty, and the minimax concave penalty (MCP) (Breheny & Huang, 2011). The advantage of mr.ash and Nash is that the user doesn't need to specify the penalty, as the model learns the mixture $(\hat{\pi}_0, \ldots, \hat{\pi}_M)$ that best fits the data via EB. (Kim et al., 2024) proposed the concept of a shrinkage operator to properly establish the connection between EB multiple linear regression and PLR, which we depict in figure 1. We further build on this idea by suggesting that some parameterizations of Nash (detailed below) can be viewed as extensions to previous PLR methods with side information.

**Group-Based and Hierarchical Penalty**  Several approaches have been developed to modulate the penalty based on groups or hierarchical structures of the data. Examples include the Group Lasso (Yuan & Lin, 2006), which uses a penalty of the form $\lambda_1 \|\boldsymbol{b}\|_1 + \lambda_2 \sum_{k \in \mathcal{K}} \sqrt{d_k} \|\boldsymbol{b}_k\|_2$, and the IPF-Lasso Boulesteix et al. (2017) with a penalty of the form $\lambda \sum_{k \in \mathcal{K}} \sum_{j \in k} \omega_k |b_j|$, where $\mathcal{K}$ corresponds to the different groups or clusters. These cases are easily handled by Nash, as they simply correspond to fitting an ash prior per group/cluster/category. This is achieved using a prior of the form $g_k(\cdot) = \pi_{0k} \delta_0 + \sum_{m=1}^{M} \pi_{mk} N(0, \sigma_m^2)$ for each $k$. In other words the side information $\boldsymbol{d}_j$ for the covariate $\boldsymbol{x}_j$ is a vector of length $\mathcal{K}$ with binary entries, where the $k^{\text{th}}$ entry of $\boldsymbol{d}_j$ is set to 1 if covariate $j$ belongs to group $k$. Thus, the model $\pi : \boldsymbol{d}_j \to (\pi_0(\boldsymbol{d}_j), \ldots, \pi_M(\boldsymbol{d}_j))$ is a multinomial regression that is straightforward to fit using standard machine learning routines. Unlike the Group Lasso or the IPF-Lasso, Nash can naturally fit different penalty types to different groups ( e.g., fitting an $L_1$ like penalty on group 1 and fitting an $L_2$ like penalty on group 2).

**Fused Lasso and Graph-Based Penalty**  The Fused Lasso (Tibshirani et al., 2005) aims to balance sparsity and smoothness covariates using a penalty of the for $\sum_{j=1}^{p} |b_j| \leq s_1$  and  $\sum_{j=2}^{p} |b_j - b_{j-1}| \leq s_2$ . Bayesian versions Casella et al. (2010); Betancourt et al. (2017) have been proposed. We extend these with graph neural networks (GNNs) to handle more complex dependencies. Classical Bayesian Fused Lasso Casella et al. (2010) can be reframed using a trivial graphical neural networks (GNN) (Kipf & Welling, 2017). Here, $\boldsymbol{d}_j = \boldsymbol{d}_j^{t+1}$ is the graph (a line in the Fused Lasso case) with nodes storing $\beta_{j,MLE}^{t+1}$ and $\bar{b}_j^t$. As the model converges, $\bar{b}_{j+1}^t \approx \bar{b}_{j+1}^{t+1}$, aligning with classic formulations. We propose the EB Fused Lasso formulation:

$$g_{\text{fused}}(\boldsymbol{d}_j) = z L(0, s_1) L(l(\boldsymbol{d}_j), s_2) L(r(\boldsymbol{d}_j), s_2) \tag{24}$$

Here, $L(\mu, s_0)$ is a Laplace distribution centered at $\mu$ with scale $s_0$, and $z$ is a normalization constant. Functions $r(\boldsymbol{d}_j) = \bar{b}_{j-1}^t$ and $l(\boldsymbol{d}_j) = \bar{b}_{j+1}^t$ are trivial GNNs, allowing different strengths for previous and next values. Posterior moments for $b_j$ can be approximated via Gauss-Hermite quadrature. Hyperparameters $(s_1, s_2)$ are learned by maximizing the marginal log likelihood.

For an arbitrary graph, model 24 becomes computationally challenging as computing the posterior under a product of $k > 3$ Laplace distributions, as it quickly becomes computationally demanding to approximate. We propose **a generalized EB Fused Lasso** :

$$g_{\text{fused}}(\boldsymbol{d}_j) = z L(0, s_1) L(v_1(\boldsymbol{d}_j), s_2(\boldsymbol{d}_j)) \tag{25}$$

Here, $v_1(\boldsymbol{d}_j), (\boldsymbol{d}_j)$ is the output of a GNN output controlling $b_j$'s smoothness with respect to the graph structure. This simplifies normalization computation and integral approximation as it only uses two Laplace distributions.

Note that different variations of the Fused Lasso have been proposed, such as the Sparse Regression Incorporating Graphical Structure Among Predictors (SRIG) Yu et al. (2016) or the Graph-Guided Fused Lasso (GGFL) Chen et al. (2010). The SRIG and GGFL penalties can also be mimicked by adapting the prior 72 using Normal instead of Laplace.

**Beyond Regularization**  We also provide an implementation of Nash that uses penalties based on Mixture Density Networks (MDN) (Bishop, 1994) and Graph Mixture Density Networks (GMDN) (Errica et al., 2021). The formulation is as follows:

$$g(j, \boldsymbol{\theta}) = \pi_0(\boldsymbol{d}_j)\delta_0 + \sum_k \pi_k(\boldsymbol{d}_j)N(\mu_k(\boldsymbol{d}_j), \sigma_k^2(\boldsymbol{d}_j)) \tag{26}$$

Here, $(\pi_k(\boldsymbol{d}_j)), (\mu_k(\boldsymbol{d}_j)), (\sigma_k^2(\boldsymbol{d}_j))$ are the outputs of the (graph) MDN, as described in Bishop (1994) and in Errica et al. (2021). These parameterizations allow the model to actually push the values of $b_j$ away from 0. Enabling Nash to be used as a "self-supervised" biased regression where the bias (here $\mu_k(\boldsymbol{d}_j)$) is automatically learned from the data.

## 5 NUMERICAL EXPERIMENT

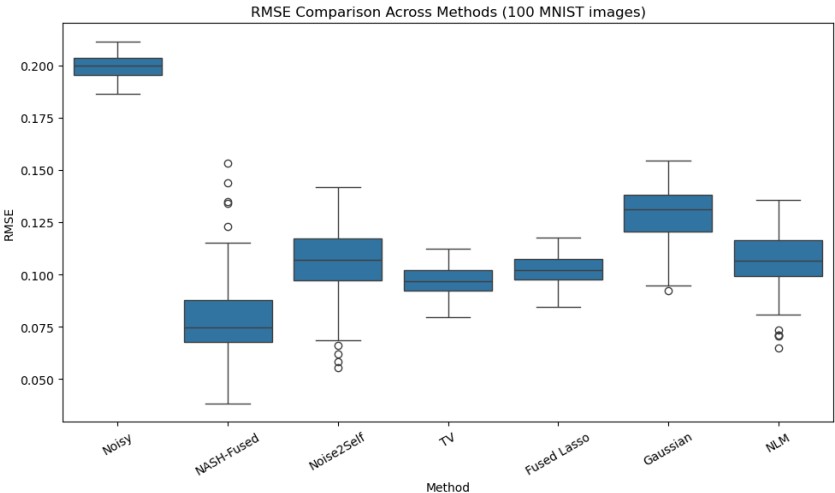

Figure 2: Performances of the different approaches for denoising MNIST image in terms of RMSE.

We evaluate Nash prediction performance on 4 real data sets that have side information. In each of these datasets, we also benchmark the performance of mr.ash, Lasso, the elastic net (Enet) with $\alpha = 0.5$, ridge regression, and when the data display group/hierarchical side information, we also benchmark the ipf Lasso. We benchmark 2 versions of Nash: i) Nash without side information and ii) Nash-mdn equation 26. We also benchmark the performance of xgboost (Chen & Guestrin, 2016), and multi-layer perceptron (MLP) with L2 regularization.

We selected a range of real datasets that spans from small data set and with a limited number of covariates (e.g. Air Passenger data ) to larger data set scale such as epigenetic age prediction with nearly 500,000 predictors. We detail these datasets and the preprocessing in A.3. In most of our experiments, we proceed as follows: we remove at random 20% of the data for testing purposes, run the different methods on the remaining data, and evaluate the performance of each method in terms of root mean squared error (RMSE).

- **SNP500**: we used daily return from of the **AAPL** symbols from SNP500 using other assets daily return. For each asset used in the predictor, we obtained the type of industry in which this asset is part of (e.g., Technology, Communication Services, Healthcare, Equity Funds ) and used it as side information.

- **Airpassenger**: we added noise to the Airpassenger data set and used the measurement time point as side information.

- **GSE40279**: we used methylation data from individuals measurement ($p = 489,503$) to predict the age of the subject (Bohlin et al., 2016; Horvath & Raj, 2018). We use methylation probe annotation as side information.

- **TCGA**: we predict individual normalized *BRCA1* gene expression (an important gene in breast cancer) expression level using the other genes. We use gene pathways from KEGG pathway as side information.

| Method | SNP500 | Airpassenger | GSE40279 | TCGA |
|---|---|---|---|---|
| Dimension ($n \times p$) | $235 \times 85$ | $144 \times 144$ | $679 \times 489{,}503$ | $1{,}212 \times 18{,}300$ |
| Side info | group | time | probe type | pathway |
| Ridge | 0.070 (0.065; 0.076) | 33.0 (31.0; 35.0) | 7.21 (6.83; 7.58) | 0.536 (0.472; 0.599) |
| Enet | 0.071 (0.066; 0.078) | 30.2 (29.5; 30.9) | 5.28 (5.02; 5.56) | 0.466 (0.411; 0.522) |
| Lasso | 0.093 (0.087; 0.100) | 49.2 (48.7; 49.7) | 5.39 (5.08; 5.71) | 0.465 (0.412; 0.518) |
| mr.ash | 0.082 (0.076; 0.088) | 20.2 (19.1; 21.3) | 5.25 (5.01; 5.71) | 0.449 (0.405; 0.493) |
| XGBoost | 0.062 (0.053; 0.069) | 18.2 (17.2; 19.1) | 6.17 (5.83; 6.52) | 0.549 (0.494; 0.604) |
| MLP | 0.441 (0.361; 0.521) | 59.5 (58.3; 60.8) | 13.62 (13.08; 14.15) | 0.457 (0.349; 0.566) |
| ipf-Lasso | 0.066 (0.060; 0.072) | NA | **5.06** (4.81; 5.33) | 0.443 (0.393; 0.473) |
| Nash.no.cov | 0.084 (0.079; 0.089) | 19.7 (18.6; 20.7) | 5.27 (5.01; 5.53) | 0.457 (0.412; 0.504) |
| Nash.mdn | **0.058** (0.053; 0.0643) | **17.7** (17.1; 18.2) | 5.12 (4.77; 5.47) | **0.435** (0.387; 0.483) |

Table 1: Comparison of methods across datasets using RMSE. Parentheses denote 95% confidence intervals based on Gaussian approximations. ipf-Lasso cannot handle time as side information, so we put NA for the Airpassenger experiment.

## 5.1 Denoising MNIST Images

We evaluate the performance of Nash-fused 24 using a simple 2-layer message passing GNN to remove Gaussian noise in images compared to methods for image denoising that only use a **single image** (as opposed to models trained on other images like diffusion model), such as fused-lasso (Tibshirani et al., 2005), Total Variation (TV) Denoising Chambolle (2004), Non-Local Means (NLM) ) (Buades et al., 2005), Gaussian Filtering (Gonzalez & Woods, 2002), Median Filtering (Huang et al., 1979), and Noise2Self (Batson & Royer, 2019) of denoising noisy grayscale images from the MNIST dataset. Nash-fused was run treating each image as a 2D grid graph, where each pixel is a node connected to its 4-nearest neighbors, and Nosie2Self was run using a convolutional neural net, which is substantially slower than Nash-fused. The true signal is the clean MNIST digit image scaled to [0,1], and additive Gaussian noise with as standard deviation of $\sigma = 0.2$ was applied to produce the observed noisy image. The experiment is repeated over 100 randomly selected MNIST images. For each method, we report the root mean squared error (RMSE) between the denoised image and the ground truth. Results are summarized as boxplots in Figure 2 additional experiments using convolutional neural nets are presented in supplementary material in Figure 3. Examples of denoised images using Nash-fused in the Appendix (see figure5-11)

## 6 Discussion

We proposed Nash, a novel high-dimensional regression framework that integrates covariate-specific side information into the estimation process using neural networks. Nash adaptively learns structured penalties in a nonparametric fashion, enabling flexible regularization without the need for cross-validation. Our method generalizes and extends existing approaches that incorporate side information, offering a unified and more expressive framework. We also proposed a new learning algorithm, split variational empirical Bayes (split VEB), which decouples prior learning from posterior inference, allowing for efficient and scalable optimization. This algorithm naturally connects to and simplifies a recently proposed variational Empirical Bayes approach(Kim et al., 2024), while supporting far richer prior families, including those parameterized by deep neural networks.

To our knowledge, Nash is the first regression method that models the prior distribution over regression effects as a direct function of side information, enabling automatic, data-driven regularization across diverse structures such as groups, time, and graphs. Through extensive experiments on real and synthetic datasets, we demonstrated that Nash consistently performs competitively and can outperform existing methods tailored to handle a **specific** type of side information.

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

# A  APPENDIX

## A.1  DETAILED SPLIT VEB FOR THE NASH MODEL

The Nash model with side information can be written as:

$$y|X, \beta, \sigma^2 \sim N(X\beta, \sigma^2) \tag{27}$$

$$\beta_j \sim N(b_j, \sigma_0^2) \tag{28}$$

$$b_j \sim g(.; d_j, \theta) \tag{29}$$

As noted in our manuscript we restrict our search to posterior of the form

$$q(\beta, b) = \prod_j^P q_{\beta_j}(\beta_j) q_{b_j}(b_j) \tag{30}$$

The overall evidence lower bound (ELBO) for the Nash model is

$$F(q_\beta, q_b, g; \sigma^2, \sigma_0^2)_{Nash} = \sum_i \mathbb{E} \log \frac{p(y_i|x_i, \beta, \sigma^2)}{q_\beta(\beta)} + \sum_j \mathbb{E} \log p(\beta_j|b_j, \sigma_0^2) + \tag{31}$$

$$\sum_j \mathbb{E} \log \frac{g(b_j; d_j, \theta)}{q_{b_j}(b_j)} \tag{32}$$

Our coordinate-ascent algorithm iterates between the updating $q_\beta$ and updating $(q_b, g(.; ., \theta))$.

### A.1.1 UPDATE FOR $q_{\beta_j}$

Note that given that when $q_{\boldsymbol{b}}, g(.;., \theta)$ are fixed then the ELBO Nash model for $q_{\beta_j}$ $j = 1, \ldots, P$ is

$$F(q_{\beta_j})_{Nash} = \mathbb{E}\log(p(\boldsymbol{y}|\boldsymbol{x}_j, \beta_j, , \boldsymbol{X}_{-j}, \boldsymbol{\beta}_{-j}, \sigma^2) + \mathbb{E}\log p(\beta_j|\bar{b}_j, \sigma_0^2) - \mathbb{E}\log q_{\beta_j} \quad (33)$$

$$= \mathbb{E}\log(p(\bar{\boldsymbol{r}}_j|\boldsymbol{x}_j, \beta_j, \sigma^2) + \mathbb{E}\log p(\beta_j|\bar{b}_j, \sigma_0^2) - \mathbb{E}\log q_{\beta_j} \quad (34)$$

where $\bar{\boldsymbol{r}}_j = \boldsymbol{y} - \boldsymbol{X}_{-j}\bar{\boldsymbol{\beta}}_{-j}$, this is direct consequence of Proposition 1 of Kim et al. (2024)), so the and so $q_{\beta_j}^* = maxF(q_{b_j})$ is given by computing the posterior of the following simple model

$$\bar{\boldsymbol{r}}_j = \boldsymbol{x}_j\beta_j + \varepsilon \quad (35)$$

$$\beta_j \sim \mathcal{N}(\bar{b}_j, \sigma_0^2) \quad (36)$$

$$\varepsilon \sim \mathcal{N}(0, \sigma^2) \quad (37)$$

By conjugacy, the posterior distribution of $\beta_j$ has the following form $\beta_j \mid \bar{\boldsymbol{r}}_j, \boldsymbol{x}_j \sim \mathcal{N}(\bar{\beta}_j, s_j^2)$ with posterior variance $s_j^2 = \left(\frac{\boldsymbol{x}_j^t\boldsymbol{x}_j}{\sigma^2} + \frac{1}{\sigma_0^2}\right)^{-1}$ and posterior mean $\bar{\beta}_j = s_j^2 \left(\frac{\boldsymbol{x}_j^t\bar{\boldsymbol{r}}_j}{\sigma^2} + \frac{\bar{b}_j}{\sigma_0^2}\right)$. In practice given that the column of $\boldsymbol{X}$ are centered $\boldsymbol{x}_j^t\boldsymbol{x}_j^t = n - 1$ for al $j = 1, \ldots, p$.

### A.1.2 UPDATE FOR $q_b$ AND $g$

Given $q_{\boldsymbol{\beta}}$ and $\sigma^2$, the ELBO for the Nash model

$$F(q_{\boldsymbol{b}}, g(.;., \theta))_{Nash} = \sum_j \mathbb{E}\log p(\bar{\beta}_j|b_j, \sigma_0^2) + \sum_j \mathbb{E}\log \frac{g(b_j; \boldsymbol{d}_j, \theta)}{q_{b_j}(b_j)} \quad (38)$$

This ELBO corresponds to a so-called (covariate)moderated normal mean problem (see (Stephens, 2017; Willwerscheid et al., 2024)) that we detail below

**The cEBNM problem** Given $p$ observations $\bar{\beta}_j \in \mathbb{R}$ with known standard deviations $s_j^2 > 0$, $j = 1, \ldots, p$, the normal means model (Stephens, 2017) is

$$\bar{\beta}_j \stackrel{\text{ind.}}{\sim} N(b_j, \sigma_0^2), \quad (39)$$

where the "true" means $b_j \in \mathbb{R}$ are unknown. We further assume that

$$b_j \stackrel{\text{i.i.d.}}{\sim} g \in \mathcal{G}, \quad (40)$$

where $\mathcal{G}$ is some prespecified family of probability distributions. The empirical Bayes (EB) approach to fitting this model exploits the fact that the noisy observations $\bar{\beta}_j$, contain not only information about the underlying means $b_j$ but also about how the means are collectively distributed (i.e., $g$). EB approaches "borrow information" across the observations to estimate $g$, typically by maximizing the marginal log-likelihood. The unknown means $b_j$ are generally estimated by their posterior mean.

We adapt EBNM to a covariate-moderated setting (covariate moderated EBNM, cEBNM), where we allow the prior for the $j$-th unknown mean to depend on additional data $\boldsymbol{d}_j$,

$$b_j \stackrel{\text{ind.}}{\sim} g(\boldsymbol{d}_j, \boldsymbol{\theta}) \in \mathcal{G}, \quad (41)$$

so that each combination of $\boldsymbol{\theta}$ and $\boldsymbol{d}_j$ maps to an element of $\mathcal{G}$. We refer to this modified EBNM model as "covariate-moderated EBNM" (cEBNM).

Solving the cEBNM problem, therefore, involves two key computations:

**1. Estimate the model parameters.** Compute

$$\hat{\boldsymbol{\theta}} := \underset{\boldsymbol{\theta} \in \mathbf{R}^m}{\arg\max} \mathcal{L}(\boldsymbol{\theta}), \quad (42)$$

where $\mathcal{L}(\boldsymbol{\theta})$ denotes the marginal likelihood,

$$\mathcal{L}(\boldsymbol{\theta}) := p(\bar{\boldsymbol{\beta}} \mid \boldsymbol{s}, \boldsymbol{\theta}, \mathbf{D}) = \prod_{j=1}^{p} \int \mathcal{N}(\bar{\beta}_j; b_j, \sigma_0^2) \, g(b_j; \boldsymbol{d}_j, \boldsymbol{\theta}) \, db_j, \tag{43}$$

in which $\bar{\boldsymbol{\beta}} = (\bar{\beta}_1, \ldots, \bar{\beta}_p)$, $\boldsymbol{s} = (s_1, \ldots, s_n)$, $\mathbf{D}$ is a matrix storing $\boldsymbol{d}_1, \ldots, \boldsymbol{d}_p$, and $\mathcal{N}(\bar{\beta}_j; b_j, \sigma_0^2)$ denotes the density of $\mathcal{N}(b_j, \sigma_0^2)$ at $\bar{\beta}_j$, and $g(b_j; \boldsymbol{d}_j, \boldsymbol{\theta})$ denotes the density of $g(\boldsymbol{d}_j, \boldsymbol{\theta})$ at $b_j$.

**2. Compute posterior summaries.** Compute summaries from the posterior distributions, such as the posterior means $\bar{b}_j := \mathbb{E}[b_j \mid \bar{\beta}_j, s_j, \hat{\boldsymbol{\theta}}, \mathbf{D}]$, using the estimated prior,

$$p(b_j \mid \bar{\beta}_j, s_j, \bar{\boldsymbol{\theta}}, \mathbf{D}) \propto \mathcal{N}(\hat{\beta}_j; b_j, \sigma_0^2) \, g(b_j; \boldsymbol{d}_j, \hat{\boldsymbol{\theta}}). \tag{44}$$

In summary, solving the cEBNM problem consists of finding a mapping from known quantities $(\bar{\boldsymbol{\beta}}, \boldsymbol{s}, \mathbf{D})$ to a tuple $(\hat{\boldsymbol{\theta}}, q)$, where each $(\boldsymbol{d}_j, \hat{\boldsymbol{\theta}})$ maps to an element $g(\boldsymbol{d}_j, \boldsymbol{\theta}) \in \mathcal{G}$, and $q$ is the posterior distribution of the unobserved $\boldsymbol{b}$ given $(\bar{\boldsymbol{\beta}}, \boldsymbol{s}, \mathbf{D})$. We denote this mapping as

$$\text{cEBNM}(\hat{\boldsymbol{\beta}}, \boldsymbol{s}, \mathbf{D}) = (\hat{\boldsymbol{\theta}}, q). \tag{45}$$

Any prior family is admissible under the cEBMF framework so long as 45 is computable.

### A.1.3 UPDATE FOR $\sigma$ AND $\sigma_0^2$

The update for $\sigma$ and $\sigma_0^2$ are obtained by simply maximizing the ELBO $F(q_{\boldsymbol{\beta}}, q_{\boldsymbol{b}}, g; \sigma^2, \sigma_0^2)_{Nash}$ with respect to $\sigma$ and $\sigma_0^2$ ,i.e;

$$(\sigma^2)^* = \arg\max_{\sigma^2} F(q_{\boldsymbol{\beta}}, q_{\boldsymbol{b}}, g; \sigma^2, \sigma_0^2)_{Nash} \tag{46}$$

$$(\sigma_0^2)^* = \arg\max_{\sigma_0^2} F(q_{\boldsymbol{\beta}}, q_{\boldsymbol{b}}, g; \sigma^2, \sigma_0^2)_{Nash} \tag{47}$$

It turns out that using results from (Wang & Stephens, 2021) and Kim et al. (2024) that the update for

$$\sigma^2 = \frac{||\boldsymbol{y} - \boldsymbol{X}\boldsymbol{\beta}|| + \boldsymbol{\beta}^t(\boldsymbol{\beta}_{MLE} - \boldsymbol{\beta}) + \sigma^2 p}{n + p} \tag{48}$$

$$\sigma_0^2 = \frac{||\boldsymbol{y} - \boldsymbol{X}\boldsymbol{b}|| + \boldsymbol{b}^t(\boldsymbol{\beta} - \boldsymbol{b}) + \sigma_0^2 p \left(1 - \frac{\sum_j \pi_0(\boldsymbol{d}_j)}{p}\right)}{n + p \left(1 - \frac{\sum_j \pi_0(\boldsymbol{d}_j)}{p}\right)} \tag{49}$$

$$\tag{50}$$

Where $\boldsymbol{\beta}_{MLE}$ is the vector of $\boldsymbol{x}_j^t \bar{\boldsymbol{r}}_j$ as defined is the section A.1.1.

---

**Algorithm 1** Algorithm for the split VEB coordinate ascent for Nash.

---

**Require:** Data $\mathbf{X} \in \mathbb{R}^{n \times p}, \mathbf{y} \in \mathbb{R}^n$; a model for , $g(., \theta)$;
1:   prior variances, $\sigma_1^2 < \cdots < \sigma_K^2$, with $\sigma_1^2 = 0$; initial estimates $\boldsymbol{\beta}, \mathbf{b}, \boldsymbol{\pi}, \sigma^2$.
2:   t=0
3: **repeat**
4:      **for** $j = 1$ to $p$ **do**
5:         $\bar{\mathbf{r}}_j \leftarrow \bar{\mathbf{r}} + \mathbf{x}_j \bar{\beta}_j^t$                                                   ▷ disregard $j$th effect
6:         $\tilde{\beta}_j^{t+1} \leftarrow \mathbf{x}_j^T \bar{\mathbf{r}}_j$                                                             ▷ OLS
7:         $\bar{\beta}_j^{t+1} \leftarrow \omega_t \tilde{\beta}_j^{t+1} + (1 - \omega_t) \bar{b}_j^t$                            ▷ Posterior mean Ridge
8:         $\bar{\mathbf{r}} \leftarrow \bar{\mathbf{r}}_j - \mathbf{x}_j \bar{\beta}_j$
9:      **end for**
10:     $\hat{\boldsymbol{\theta}} = \arg\max_{\boldsymbol{\theta}} \prod_{j=1}^p \int \mathcal{N}(\bar{\beta}_j^{t+1}; b_j, \sigma_0^2) \, g(b_j; \boldsymbol{d}_j, \boldsymbol{\theta}) \, db_j$               ▷ EB for penalty
11:     $\boldsymbol{b}^{t+1} \leftarrow p(\boldsymbol{b}|\bar{\boldsymbol{\beta}}, \boldsymbol{D}, \sigma_0^2, \hat{\theta})$                                 ▷ Update latent space
12:     $\sigma_{t+1}^2 = \arg\max_{\sigma^2} F(q_{\boldsymbol{\beta}}^{t+1}, q_{\boldsymbol{b}}^{t+1}, g^{t+1}, \sigma^2, \sigma_{0t}^2)_{\text{Nash}}$
13:     $\sigma_{0t+1}^2 = \arg\max_{\sigma_0^2} F(q_{\boldsymbol{\beta}}^{t+1}, q_{\boldsymbol{b}}^{t+1}, g^{t+1}, \sigma_{t+1}^2, \sigma_0^2)_{\text{Nash}}$
14:     $\omega_{t+1} = \frac{(n-1)\sigma_{0t+1}^2}{\sigma_{t+1}^2 + (n-1)\sigma_{0t+1}^2}$
15:     $t \leftarrow t + 1$
16: **until** termination criterion is met
17: **return** $\bar{\mathbf{b}}, \boldsymbol{\pi}, \sigma^2$

---

## A.2   NASH OPTIMIZES A LOWER BOUND OF MR.ASH ELBO

The derivations below are adapted from Section 4.2.1 of Xie (2023). While Xie (2023) studies a different problem (smoothing Poisson counts), we draw inspiration from this work and adapt it to the high-dimensional Gaussian setting. In this section, we show that in the absence of side information, Nash optimizes a lower bound of the mr.ash evidence lower bound (ELBO).

Recall the two models, mr.ash and Nash:

$$\text{mr.ash} \qquad\qquad\qquad \text{Nash} \qquad\qquad\qquad\qquad (51)$$

$$\boldsymbol{y}|\boldsymbol{X}, \boldsymbol{b}, \sigma^2 \sim N(\boldsymbol{X}\boldsymbol{b}, \sigma^2) \qquad\qquad \boldsymbol{y}|\boldsymbol{X}, \boldsymbol{\beta}, \sigma^2 \sim N(\boldsymbol{X}\boldsymbol{\beta}, \sigma^2) \qquad (52)$$

$$b_j \sim g(.) \qquad\qquad\qquad \beta_j \sim N(b_j, \sigma_0^2) \qquad\qquad (53)$$

$$b_j \sim g(.) \qquad\qquad (54)$$

We consider the case without side information, recall that both approach restrict their search to posteriors of the form

$$\text{mr.ash} \qquad\qquad\qquad \text{Nash} \qquad\qquad\qquad\qquad (55)$$

$$q(\boldsymbol{b}) = \prod_j^P q_{b_j}(b_j) \qquad\qquad q(\boldsymbol{\beta}, \boldsymbol{b}) = \prod_j^P q_{\beta_j}(\beta_j) q_{b_j}(b_j) \qquad (56)$$

So the corresponding ELBO for the two models are

$$F(q_{\boldsymbol{\beta}}, g; \sigma^2)_{mr.ash} = \sum_i \mathbb{E} \log \frac{p(y_i|\boldsymbol{x}_i, \boldsymbol{b}, \sigma^2)}{q_{\boldsymbol{b}}(\boldsymbol{b})} + \sum_j \mathbb{E} \log \frac{g(b_j)}{q_{b_j}(b_j)} \qquad (57)$$

$$F(q_{\boldsymbol{\beta}}, q_{\boldsymbol{b}}, g; \sigma^2 \sigma_0^2)_{Nash} = \sum_i \mathbb{E} \log \frac{p(y_i|\boldsymbol{x}_i, \boldsymbol{\beta}, \sigma^2)}{q_{\boldsymbol{\beta}}(\boldsymbol{\beta})} + \sum_j \mathbb{E} \log p(\beta_j|b_j, \sigma_0^2) + \sum_j \mathbb{E} \log \frac{g(b_j)}{q_{b_j}(b_j)}$$
$$(58)$$

Below we show that the profiled ELBO $F(q_\beta, g; \sigma^2)_{Nash} = max_{q_{\boldsymbol{b}}} F(q_{\boldsymbol{\beta}}, q_{\boldsymbol{b}}, g; \sigma^2)_{Nash}$ is a lower bound for $F(q_\beta, g; \sigma^2)_{mr.ash}$.

### A.2.1 ELBO FOR $b_j$ IN THE NASH MODEL

The introduction of latent variable $\beta_j$ induces a marginal density of $b_j$ as

$$\log p(\boldsymbol{y}|\boldsymbol{x}, b_j, \sigma^2, \sigma_0^2) = \log \int p(\boldsymbol{y}|\boldsymbol{X}, \beta_j, \sigma^2) N(\beta_j, b_j, \sigma_0^2) g(b_j) d\beta_j \tag{59}$$

We denote $\log p(\boldsymbol{y}|\boldsymbol{x}, b_j, \sigma_0^2)$ by $\log f(b_j)$. Before demonstrating that Nash optimizes a lower bound for the mr.ash approximation, we first introduce a lemma.

**Lemma** *The second order derivative of $\log f(\cdot)$ with respect to b is lower bounded by $-1/\sigma_0^2$.*

*Proof.* The second derivative of $\log f(b)$ is

$$\frac{d^2 \log f(b_j)}{db_j^2} = \frac{f''(b_j)}{f(b_j)} - \left(\frac{f'(b_j)}{f(b_j)}\right)^2 \tag{60}$$

where

$$f'(b_j) = \frac{1}{\sigma_0^2} f(b_j) \int \beta p(\beta \mid b_j) g(b_j) d\beta - \frac{\beta}{\sigma_0^2} f(b_j) = \frac{1}{\sigma_0^2} f(b_j) \left(\mathbb{E}(\beta) - b_j\right),$$

$$f''(b_j) = \frac{1}{(\sigma_0^2)^2} f(b_j) \left(\mathbb{E}(\beta^2) - b_j \mathbb{E}(\beta)\right) - \frac{1}{\sigma_0^2} f(b_j) - \frac{b_j}{\sigma_0^2} \tag{61}$$

$$= f(b_j) \left(\frac{1}{(\sigma_0^2)^2} \mathbb{E}(\beta^2) - \frac{2b_j}{(\sigma_0^2)^2} \mathbb{E}(\beta) - \frac{1}{\sigma^2} + \frac{b_j^2}{(\sigma_0^2)^2}\right) \tag{62}$$

where the expectation are under $p(\beta_j|y, \boldsymbol{X}, b_j, \sigma, \sigma_0^2)$

Substituting $f'(b_j)$ and $f''(b_j)$, we have

$$\frac{d^2 \log f(b_j)}{db_j^2} = -\frac{1}{\sigma_0^2} + \frac{1}{(\sigma_0^2)^2} (\mathbb{E}(\beta^2) - \mathbb{E}((\beta)^2) \geq -\frac{1}{\sigma_0^2} \tag{63}$$

The primary objective is to perform inference on $b_j$, the most straightforward approach is to regard the marginal distribution as the prior of $b_j$ and maximize the corresponding ELBO,

$$\tilde{F}(q_{\beta_j}, \sigma^2) = \mathbb{E} \log \frac{p(\boldsymbol{y}|\boldsymbol{X}, \beta_j, \boldsymbol{\beta}_{-j})}{q_{\beta_j}} + \mathbb{E} \log f(\beta_j; g, \sigma^2). \tag{9}$$

The following theorem shows that the ELBO function maximized by the splitting approach is a lower bound of $F(q_{\beta_j}; \sigma^2)$.

**Theorem A.1.** *The profiled ELBO function $F(q_{b_j}; \sigma^2, \sigma_0^2) = \max_{q_{\beta_j}} F(q_{\beta_j}, q_{b_j}; \sigma^2, \sigma_0^2)_{Nash}$ is a lower bound of $F(q_{b_j}; \sigma^2)_{mr.ash}$.*

*Proof.* The ELBO of Nash for $\beta_j, b_j$ is

$$F(q_{\beta_j}, q_{b_j}; \sigma^2, \sigma_0^2)_{Nash} = \mathbb{E} \log p(\boldsymbol{y}|\boldsymbol{X}, \beta_j, \boldsymbol{\beta}_{-j}) + \mathbb{E} \log \frac{N(\beta_j, \bar{b}_j, \sigma_0^2)}{q_{\beta_j}(\beta_j)} + \mathbb{E} \log \frac{g(b_j)}{q_{b_j}} - \frac{V_{q_{b_j}}}{2\sigma_0^2} \tag{64}$$

$$= \mathbb{E} \log p(\boldsymbol{r}_j|\boldsymbol{X}_j, \beta_j) + \mathbb{E} \log \frac{N(\beta_j, \bar{b}_j, \sigma_0^2)}{q_{\beta_j}(\beta_j)} + \mathbb{E} \log \frac{g(b_j)}{q_{b_j}} - \frac{V_{q_{b_j}}}{2\sigma_0^2} \tag{65}$$

Where $\boldsymbol{r}_j = \boldsymbol{y} - \boldsymbol{X}_{-j}\boldsymbol{\beta}_{-j}$, $\bar{b}_j = \mathbb{E}_{q_{b_j}}(b_j)$ and $V_{q_{b_j}} = \mathbb{E}_{q_{b_j}}(b - \bar{b})$

Therefore the profiled Nash ELBO for $q_{b_j}, g$ is

$$F(b_j, g; \sigma^2, \sigma_0^2) = max_{q_{\beta_j}} F(q_{\beta_j}, q_{b_j}; \sigma^2, \sigma_0^2)_{Nash} \tag{66}$$

$$= \log p(\boldsymbol{r}_j|\bar{b}_j, \sigma^2) + \mathbb{E} \log \frac{g(b_j)}{q_{b_j}} - \frac{V_{q_{b_j}}}{2\sigma_0^2} \tag{67}$$

The ELBO $F(q_{\beta_j}, q_{b_j}; \sigma^2, \sigma_0^2)_{Nash}$ reach its maximum over $q_{\beta_j}$ at $q_{\beta_j}^* = p(\beta_j | r_j, \bar{b}_j, \sigma_0^2)$

A second order Taylor series expansion of $F(q_{b_j}, g; \sigma^2)_{mr.ash}$ in around $\bar{b}_j$ gives

$$F(q_{b_j}; \sigma^2)_{mr.ash} = \mathbb{E} \log p(\boldsymbol{y} | \boldsymbol{X}, b_j, \boldsymbol{b}_{-j}, \sigma^2, \sigma_0^2) + \mathbb{E} \log \frac{g(b)}{q_b} \tag{68}$$

$$= \log p(\boldsymbol{r}_{-j} | b_j, \sigma^2, \sigma_0^2) + \frac{1}{2} \left( \frac{d^2 f(b)}{db^2} \right) \bigg|_{b=\bar{b}} V_{q_b} + \mathbb{E} \log \frac{g(b)}{q_b} \tag{69}$$

$$\geq \log p(\boldsymbol{r}_{-j} | b_j, \sigma^2, \sigma_0^2) - \frac{1}{2\sigma_0^2} V_{q_b} + \mathbb{E} \log \frac{g(b)}{q_b} \tag{70}$$

$$= max_{q_{\beta_j}} F(q_{\beta_j}, q_{b_j}; \sigma^2, \sigma_0^2)_{Nash} \tag{71}$$

where $\theta$ is between $\hat{\beta}$ and $\beta$. The first inequality holds due to the Lemma above, and the second inequality is due to the definition of ELBO. $\qquad \square$

### A.3 REAL DATA EXPERIMENT

#### A.3.1 GROUP-BASED AND HIERARCHICAL SIDE INFORMATION

This section clarifies the construction of $(X, y)$, the definition of the side information $d_j$, the neural architecture used to parameterize the prior, and how the prior induces covariate-specific shrinkage. This applies to the datasets *SNP500*, *GSE40279*, and *TCGA*.

**Side information.** In group-structured regression, each covariate $j$ belongs to a group $k(j) \in \{1, \ldots, K\}$. We encode this using a one-hot vector.

$$d_j \in \{0, 1\}^K, \qquad d_{j,k} = 1 \iff j \in \text{group } k.$$

This representation allows the prior on coefficient $b_j$ to depend explicitly on the group membership of covariate $j$.

**Prior parameterization.** We use a mixture density network (MDN) to produce the group-specific shrinkage prior

$$g(b_j \mid d_j, \theta) = \pi_0(d_j)\, \delta_0 \;+\; \sum_{k=1}^{M} \pi_k(d_j)\, N\big(\mu_k(d_j),\, \sigma_k^2(d_j)\big),$$

with $g_0 = \delta_0$ fixed. The MDN learns the smooth mapping $d_j \mapsto \{\pi_k(d_j), \mu_k(d_j), \sigma_k^2(d_j)\}$ so that different groups receive different shrinkage patterns.

**Neural network architecture.** The MDN is implemented as a fully connected feed-forward network with three affine layers and ReLU activations. Given $d_j \in \mathbb{R}^q$, the network computes

$$h_1 = \text{ReLU}(W_1 d_j + b_1), \qquad h_2 = \text{ReLU}(W_2 h_1 + b_2),$$

$$\pi(d_j; \theta) = \text{Softmax}(W_3 h_2 + b_3).$$

Here,

$$W_1 \in \mathbb{R}^{H \times q}, \quad W_2 \in \mathbb{R}^{H \times H}, \quad W_3 \in \mathbb{R}^{(M+1) \times H},$$

and we use $H = 32$ in all experiments. The final layer outputs the $(M + 1)$ mixture weights for the spike-and-slab prior, along with $2M$ linear outputs for the means and log-variances of the continuous mixture components.

All networks in this section are trained using the Adam optimizer with a learning rate $10^{-3}$ for 100 epochs.

#### A.3.2 CONTINUOUS SIDE INFORMATION

In the *AirPassengers* application, the side information associated with each coefficient is continuous rather than categorical. This section clarifies how the side information $d_j$ is constructed and how the MDN adapts shrinkage continuously across covariates.

**Side information.** When covariates are associated with continuous metadata—such as time, spatial location, or continuous annotations—we encode each covariate $j$ using a real-valued feature vector

$$d_j \in \mathbb{R}^q,$$

where $q$ depends on the application. Examples include (i) $d_j = t_j$ for time-indexed covariates, (ii) $d_j = (x_j, y_j)$ for 2D spatial structure, and (iii) multi-dimensional continuous annotations. This allows the prior strength and sparsity pattern to vary smoothly as a function of $d_j$.

**Prior parameterization via an MDN.** The mixture density network used here is identical to the architecture described in Section A.3.1. It outputs the mixture weights and mixture parameters as continuous functions of $d_j$.

All MDNs in this section are trained using Adam with a learning rate $10^{-3}$ for 100 epochs.

### A.3.3 GRAPH-BASED SIDE INFORMATION

In the MNIST denoising experiment, the covariates correspond to pixels on a two-dimensional grid. This induces a natural graph structure that the prior can exploit to encourage spatial smoothness while allowing for sharp boundaries and local adaptivity. Below we describe the construction of the side information $d_j$, the graph $G = (V, E)$, and the neural architecture used to parameterize the fused prior.

**Side information from 2D coordinates.** Each coefficient $b_j$ corresponds to a pixel located at $(d_{j_1}, d_{j_2})$ on an $n \times n$ grid. We encode its spatial position as the normalized coordinate pair

$$d_j = (d_{j_1}/n, \ d_{j_2}/n),$$

so that both components lie in $[0, 1]$. This representation allows the prior to vary smoothly across both spatial dimensions.

**Graph construction.** We equip the pixel grid with a 4-nearest-neighbor graph:

$$E = \{(j, k) : k \in \text{Nbh}(j)\}, \qquad \text{Nbh}(j) = \{\text{up}, \text{down}, \text{left}, \text{right}\}.$$

This graph encodes local spatial adjacency and enables the use of message passing to incorporate information from neighboring pixels.

**Fused-Laplace prior.** To encourage piecewise smoothness, we employ a fused-Laplace shrinkage prior of the form

$$g_{\text{fused}}(d_j) \ = \ z \, L(0, s_1) \, L(v_1(d_j), \, s_2(d_j)), \tag{72}$$

where $L(\mu, s)$ denotes a Laplace distribution with location $\mu$ and scale $s$, and $z$ is a normalizing constant. The first factor shrinks $b_j$ toward zero, while the second penalizes the local difference $v_1(d_j)$ between $b_j$ and its neighbors. Learning $(s_1, s_2(d_j), v_1(d_j))$ produces a data-adaptive analogue of the fused lasso.

**Neural parameterization via a 2-layer message-passing GNN.** The parameters of the fused prior

$$(s_1, \ v_1(d_j), \ s_2(d_j))$$

are produced by a *two-layer message-passing neural network* (MPNN) operating on the graph $G = (V, E)$. Let $h_j^{(0)} = d_j$ denote the initial node features. The GNN performs two rounds of message passing:

$$h_j^{(1)} = \text{ReLU}\left( W_1 h_j^{(0)} \ + \ \sum_{k \in \text{Nbh}(j)} W_{\text{msg}} h_k^{(0)} \right),$$

$$h_j^{(2)} = \text{ReLU}\left( W_2 h_j^{(1)} \ + \ \sum_{k \in \text{Nbh}(j)} W'_{\text{msg}} h_k^{(1)} \right).$$

A final linear layer produces the prior parameters:

$$(s_1, \ s_2(d_j), \ v_1(d_j)) = \text{Softplus}\left( W_3 h_j^{(2)} + b_3 \right).$$

This architecture allows the shrinkage strength $s_1$, the smoothness scale $s_2(d_j)$, and the fused-lasso direction $v_1(d_j)$ to depend not only on the spatial coordinates of pixel $j$, but also on the features of its neighbors through message passing.

We use $H = 64$ hidden dimensions and train all GNNs using the Adam optimizer with learning rate $10^{-3}$ for 100 epochs.

### A.3.4 OTHER METHODS

**MLP/Feed forward neural network:**   We trained a feed-forward neural network (NN) as a baseline for comparison. The network consists of three fully connected layers: an input layer followed by two hidden layers of sizes 128 and 64, each with ReLU activation. The output is a single linear unit predicting a continuous response. The input features were standardized to zero mean and unit variance based on the training set. The model was trained using the Adam optimizer with a learning rate of 0.001 and mean squared error (MSE) loss. Each model was trained for 100 epochs using all training samples in batch mode. Training and evaluation were implemented in PyTorch. RMSE was computed on each test split, and average RMSE across the 10 folds is reported.

**xgboost:**   We trained gradient-boosted decision trees using the xgboost *R* package with default hyperparameters. Specifically, we used roost mean square error as objective for regression. Each model was trained using 50 boosting iterations (default parameter) with a maximum tree depth of 6 and a learning rate (eta) of 0.3.

**MNIST**   We evaluated seven denoising methods on the MNIST dataset with additive Gaussian noise of standard deviation $\sigma = 0.2$, applied independently to each pixel. Each method was applied to 20 randomly selected test images, and performance was measured using root mean squared error (RMSE) against the clean image. The Noise2Self model was a convolutional neural network (CNN) with three convolutional layers: $\text{Conv}_{1 \to 32} \to \text{ReLU} \to \text{Conv}_{32 \to 32} \to \text{ReLU} \to \text{Conv}_{32 \to 1}$. It was trained using masked pixel regression, where approximately 10% of pixels were randomly set to zero during each training iteration and the model was trained to reconstruct them. The loss was computed only over masked pixels using mean squared error. We trained the Noise2Self model for 5 epochs using the Adam optimizer with a learning rate of $10^{-3}$ and batch size 64.

The Nash-fused was trained using a message passing GNN representing each image as a 784-node graph corresponding to a $28 \times 28$ grid, with 4-neighbor connectivity and node features consisting of the noisy intensity and normalized spatial coordinates. The underlying graph neural network had two hidden layers with ReLU activations. We trained Nash-fused separately for each image using 300 steps of gradient descent with the Adam optimizer and learning rate $10^{-2}$.

Classical baselines included total variation (TV) denoising, fused lasso, Gaussian smoothing, and non-local means (NLM). TV denoising used regularization weight 0.1. Fused lasso was formulated as a convex optimization problem with an $\ell_2$ data fidelity term and isotropic TV penalty, solved using `cvxpy` with the SCS solver. Gaussian filtering used a fixed kernel with standard deviation $\sigma = 1$. NLM used the implementation from `skimage.restoration` with parameters $h = 1.15 \cdot \hat{\sigma}$ (where $\hat{\sigma}$ is estimated from the image), patch size 3, and patch distance 5.

**CNN:**   We also included a small convolutional neural network as a baseline for image denoising. The model receives a noisy 28×28 grayscale image and predicts a clean version of the same dimensions. Its architecture consists of four convolutional layers with ReLU nonlinearities, following a standard encoder–decoder design: Conv1: 1→32 filters, kernel 3×3, padding 1, ReLU; Conv2: 32→64 filters, kernel 3×3, padding 1, ReLU; Conv3: 64→32 filters, kernel 3×3, padding 1, ReLU; Conv4: 32→1 filter, kernel 3×3, padding 1, linear output.

The model is trained via mean squared error (MSE) between predicted and true clean pixels, using Adam optimizer with a learning rate of $10^{-3}$ using five epochs, on a single image. See Figure 3

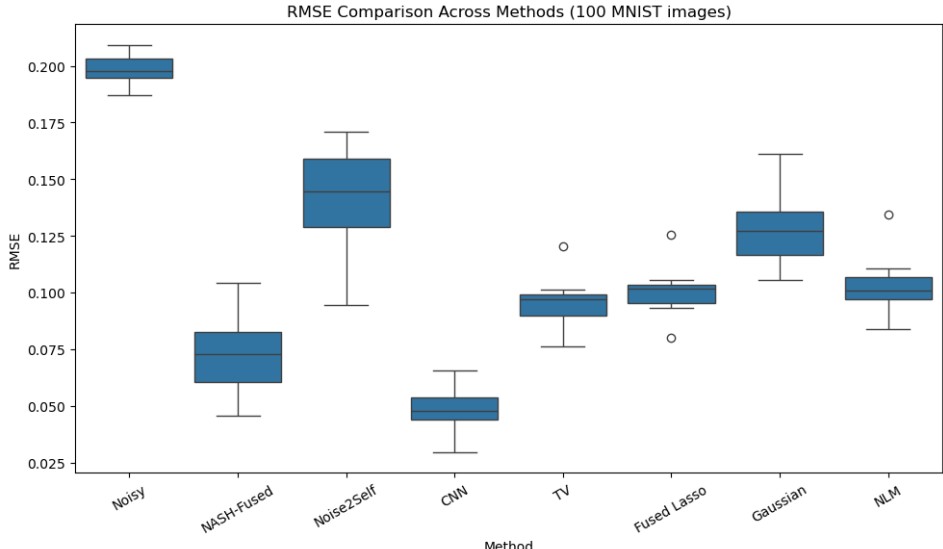

Figure 3: performances of the different approaches for denoising 100 MNIST images in terms of RMSE.

## A.4 COMPARISON WITH MR.ASH

In figure 4a, we showcase a couple of examples where we compare Nash and mr.ash both in terms of ELBO and fitted performance. The ELBO of Nash is particularly cumbersome to compute, as it requires storing a large number of parameters ($O(MP)$), where $M$ is the number of mixture components in the prior and $P$ is the number of covariates). In the example below, we display Mr.ash $ELBO_{mr.ash} + \sum_j \mathbb{E} \log \frac{p(\beta_j|b_j,\sigma_0^2)}{q_\beta(\beta)}$. In practice we monitor convergence as in mr.ash Kim et al. (2024) (and other commonly used variational method such as SuSiE-inf Cui et al. (2024)) by stopping the iteration when $||\beta^{t+1} - \beta^t||_2^2 < 1e-6$. In **Algorithm 2** below, we also provide a high-level description of the differences between Nash and mr.ash.

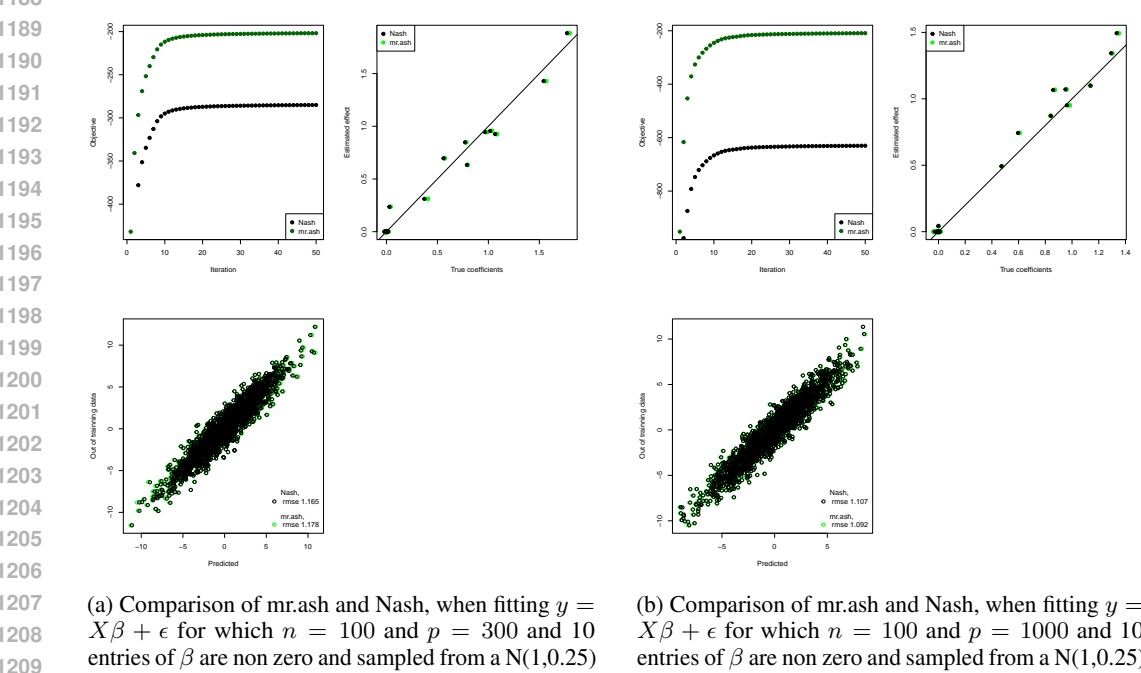

(a) Comparison of mr.ash and Nash, when fitting $y = X\beta + \epsilon$ for which $n = 100$ and $p = 300$ and 10 entries of $\beta$ are non zero and sampled from a N(1,0.25) and $\epsilon_i \sim N(0,1)$

(b) Comparison of mr.ash and Nash, when fitting $y = X\beta + \epsilon$ for which $n = 100$ and $p = 1000$ and 10 entries of $\beta$ are non zero and sampled from a N(1,0.25) and $\epsilon_i \sim N(0,1)$

Figure 4: Comparison of mr.ash and Nash in two case examples

**Algorithm 2:** Side-by-side pseudo-code comparison of the learning procedures for `mr.ash` and `Nash`. We highlight in red the main computational differences between mr.ash and Nash

| mr.ash | Nash |
|---|---|
| **Require:** $X, y$, prior model $g(\theta)$. | **Require:** $X, y$, prior model $g(\theta)$. |
| 1: **repeat** | 1: **repeat** |
| 2:      **for** $j = 1$ to $p$ **do** | 2:      **for** $j = 1$ to $p$ **do** |
| 3:         $\bar{r}_j \leftarrow \bar{r} + x_j \bar{b}_j$ | 3:         $\bar{r}_j = y - \sum_{j' \neq j} x_{j'} \bar{\beta}_{j'}$ |
| 4:         $\tilde{b}_j \leftarrow x_j^\top \bar{r}_j$ | 4:         $\tilde{\beta}_j = x_j^\top \bar{r}_j$ |
| 5:         Update $\bar{b}_j$ given $\tilde{b}_j$ | 5:         Update $\bar{\beta}_j$ given $\tilde{\beta}_j$ |
| 6:         Update $g(\hat{\theta})$ given $\tilde{b}_j$(M-step) | 6: |
| 7:         $\bar{r} \leftarrow \bar{r}_j - x_j \bar{b}_j$ | 7:         $\bar{r} \leftarrow \bar{r}_j - x_j \bar{\beta}_{j'}$ |
| 8:      **end for** | 8:      **end for** |
| 9:      Update $g(\hat{\theta})$ using $(\tilde{b}_j)_{j=1,...,P}$ (E-step) | 9:      Update $g(\hat{\theta})$ using $(\tilde{\beta}_j)_{j=1,...,P}$ |
| 10: **until** convergence | 10: **until** convergence |

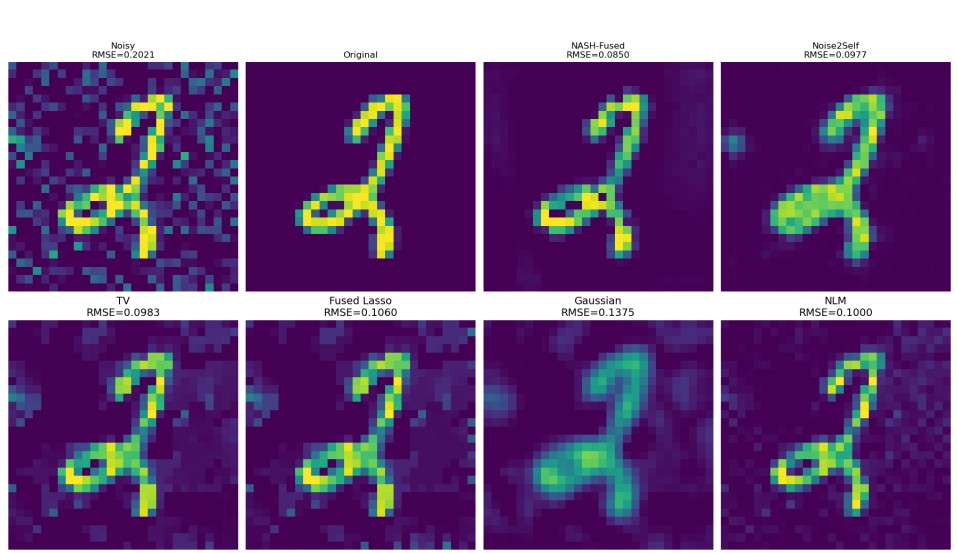

Figure 5: Additional denoised image

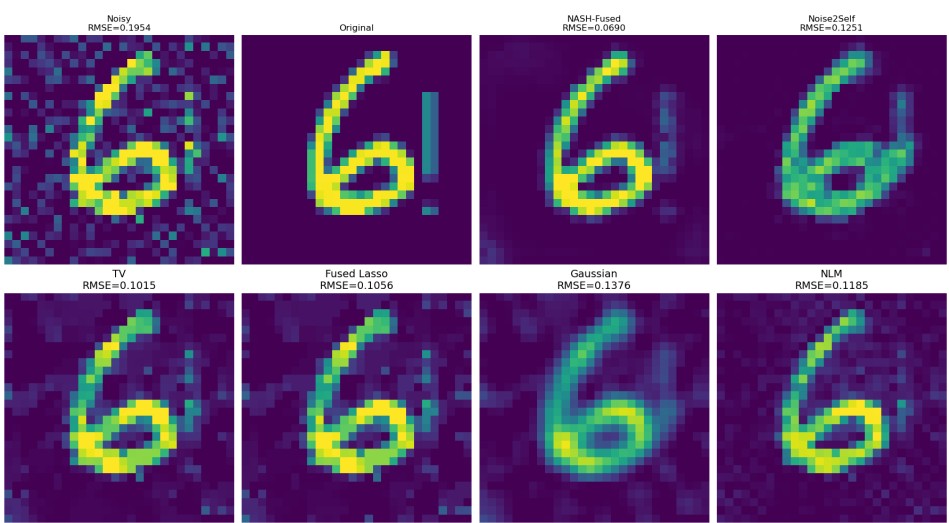

Figure 6: Additional denoised image

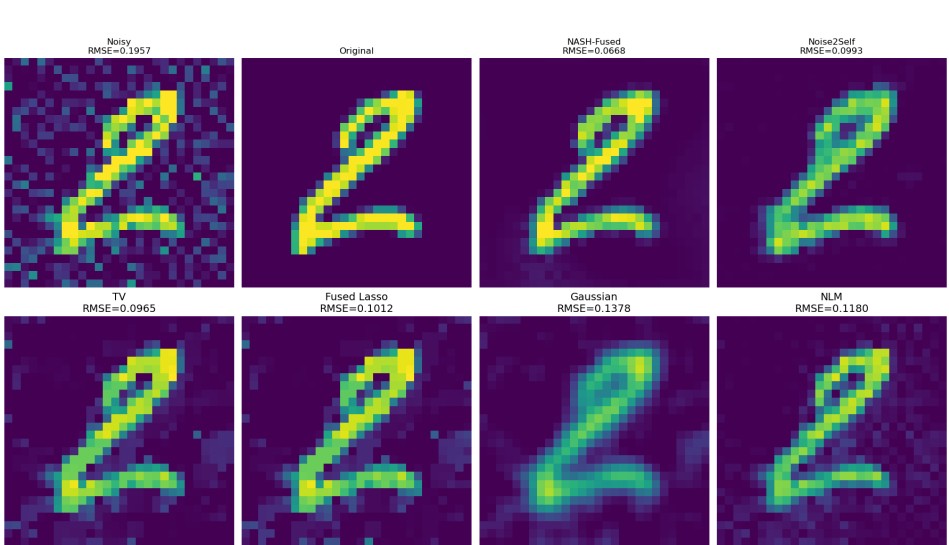

Figure 7: Additional denoised image

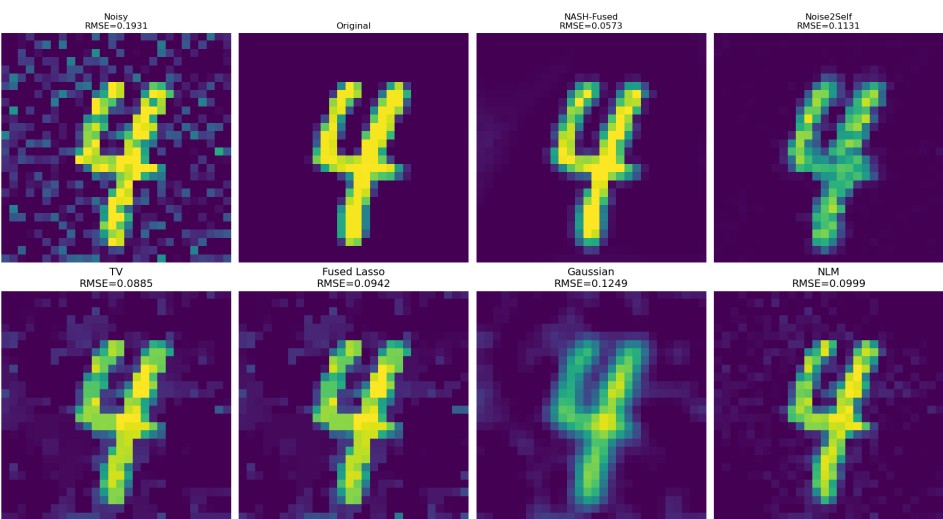

Figure 8: Additional denoised image

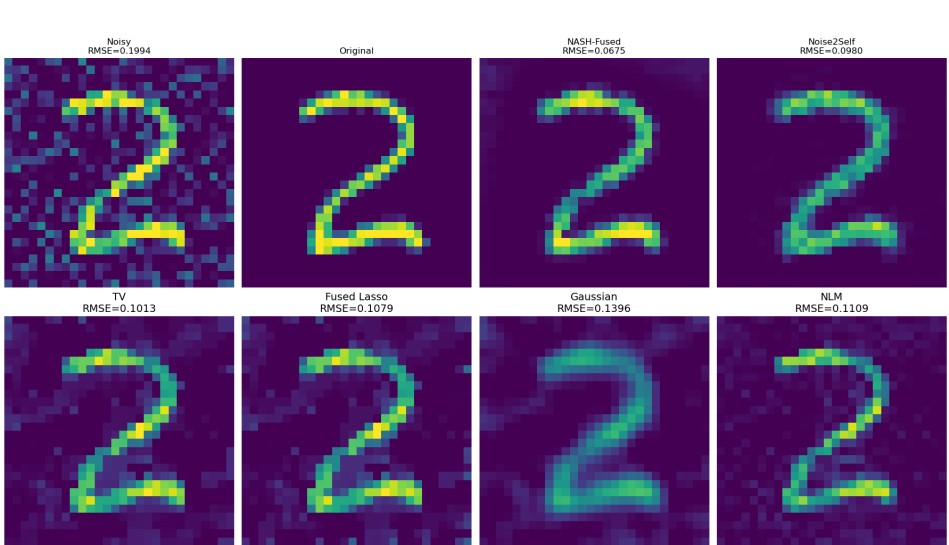

Figure 9: Additional denoised image

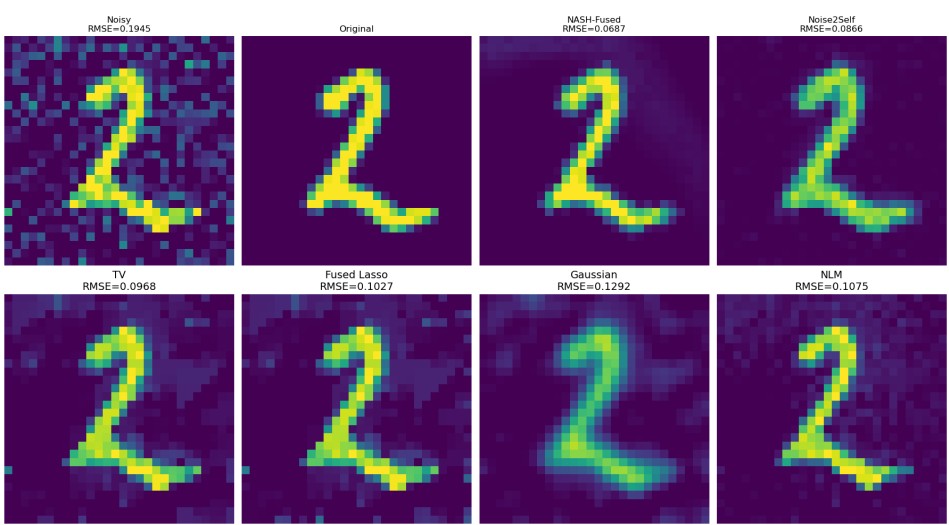

Figure 10: Additional denoised image

1404
1405
1406
1407
1408
1409
1410
1411
1412
1413
1414
1415
1416
1417
1418
1419
1420
1421
1422
1423
1424
1425
1426
1427
1428
1429
1430
1431
1432
1433
1434
1435
1436
1437
1438
1439
1440
1441
1442
1443
1444
1445
1446
1447
1448
1449
1450
1451
1452
1453
1454
1455
1456
1457

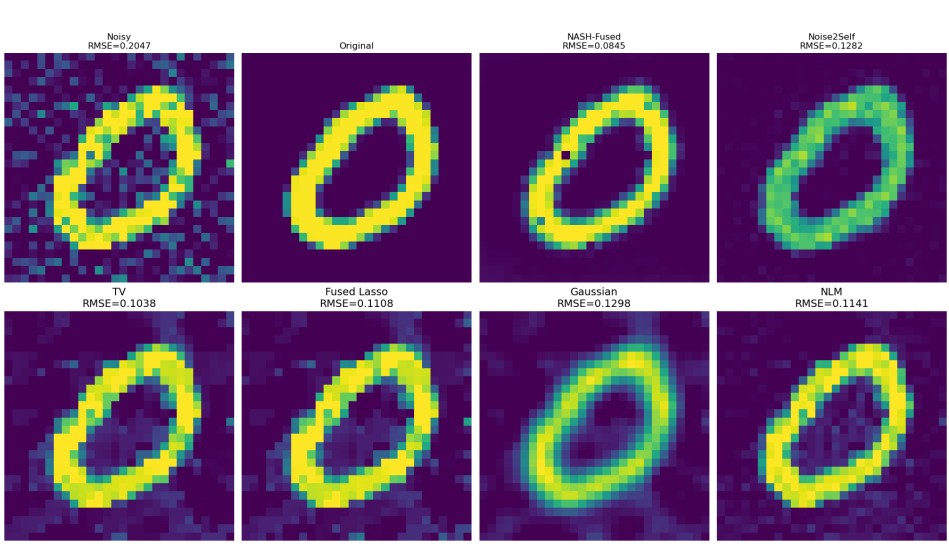

Figure 11: Additional denoised image

