# OpenReview forum: "Nash: Neural Adaptive Shrinkage for Structured High-Dimensional Regression"
_ICLR.cc/2026/Conference — ICLR 2026 Conference Withdrawn Submission_

### Official Review · Reviewer_7a2y · 2025-11-01

**Soundness:** 3
**Presentation:** 3
**Contribution:** 3
**Rating:** 6
**Confidence:** 3

**Summary:**

This paper introduces Neural Adaptive Shrinkage (NASH), a novel framework for high-dimensional regression that leverages neural networks to incorporate covariate-specific side information. The core idea is to learn an adaptive, structured penalty function on a per-covariate basis, guided by the side information. This is framed within an empirical Bayes perspective, and the authors propose an efficient "split VEB" (Variational Empirical Bayes) algorithm for model fitting, which decouples prior learning from posterior computation. The method is shown to generalize several existing penalized regression techniques and demonstrates strong empirical performance on various real-world datasets and an image denoising task.

**Strengths:**

- Provides a novel and unified method to integrate diverse side information (e.g., groups, graphs, time) into regression using a neural network-based prior.
- The proposed split VEB algorithm is scalable and efficient, making the approach practical by decoupling the prior learning and posterior computation steps.
- The method is shown to be highly competitive and often outperforms established baselines across a comprehensive set of real-world experiments.

**Weaknesses:**

- Performance can be sensitive to the neural network architecture and its hyperparameters, which adds a layer of complexity to its practical application.
- The use of a neural network to define the penalty structure may reduce the model's interpretability compared to classical regularization techniques.
- The paper is primarily empirical and would be strengthened by theoretical results, such as convergence guarantees for the proposed algorithm.

**Questions:**

Can the trained neural network provide insights into the underlying structure of the covariates? For example, is it possible to interpret what features of the side information the model found most important for determining the regularization?

---

> ### Author Response · Authors · 2025-11-19
>
> We thank the reviewer for their time and constructive feedback.
>
>
> **On interpretability and neural network-based penalties:**
> We appreciate this concern. While neural networks add parameters, this flexibility enables Nash to adapt to complex, nonlinear side information that classical methods cannot capture. Importantly, Nash maintains interpretability despite using neural networks: when using spike-and-slab priors (Equations 12, 25), we can directly visualize the learned function$
> \pi_0(d_j)$  the probability of shrinking coefficient $b_j$ to zero. Higher values indicate stronger regularization, with  $\pi_0(d_j)=1$ meaning complete exclusion of covariate j.
>
> A closely related visualization approach appears in Denault et al. ("Covariate-moderated Empirical Bayes Matrix Factorization," NeurIPS 2025), where the authors visualize learned mixture weights  $\pi_0(d_j)$ as a function of 2D spatial coordinates (their Figure 1G), providing intuitive interpretation of how side information influences shrinkage. The same principle applies directly to Nash.
>   Such visualizations provide actionable insights into how Nash adapts regularization based on side information.
>
> Additionally, standard explainability tools (Shapley values, integrated gradients) can be applied to the neural network mapping $d_j$ → $π(d_j)$ to identify which features most influence regularization decisions.
>
> **On theoretical convergence guarantees:**
> Nash employs coordinate ascent variational inference (CAVI), which is guaranteed to converge to a local optimum of the ELBO under standard regularity conditions. Each coordinate update increases the ELBO, ensuring monotonic improvement. We acknowledge that stronger theoretical results—such as global optimality conditions or finite-sample guarantees—would be valuable but remain challenging for variational methods with neural components. This represents important future work.
>
> We hope these clarifications address the reviewer’s concerns and that you may consider increasing your evaluation score.

---

### Official Review · Reviewer_fhSk · 2025-11-02

**Soundness:** 3
**Presentation:** 2
**Contribution:** 2
**Rating:** 4
**Confidence:** 3

**Summary:**

The paper introduces a new method to adaptively penalize the coefficients in linear regression. The authors provide a model, derive a loss and an algorithm, and validate on MNIST and tabular data.

**Strengths:**

The topic is interesting and the paper is rather easy to read.

**Weaknesses:**

Weaknesses:
- Clarity:
    - I am not sure I understood the proposed algorithm, would it be possible to encapsulate it in an environment, as done in [1], maybe it would be a good opportunity to highlight the difference with [1]
    - In Figure 1, bottom left, where dore the GNN-based prior comes from? Is it a pre-trained GNN? On other data? Could authors provide more details on this specific figure?
- Novelty: I am not sure I understood the difference with [1], could authors comment on that? Is it a generalization of [1] to graph-based data?
- Experiments
    - Would it be possible to add a vanilla CNN as a benchmark for Figure 2? (I understand this is a different kind of technique, just want to have an order of magnitude)
    - Table 1, how significant are the performance gain?


[1] Youngseok Kim, Wei Wang, Peter Carbonetto, and Matthew Stephens. A flexible empirical Bayes approach to multiple linear regression and connections with penalized regression. Journal of Machine Learning Research, 25

**Questions:**

see weaknesses

---

> ### Author Response · Authors · 2025-11-19
>
> We thank the reviewer for the  constructive and detailed feedback.
>
> **Clarity of the algorithm.**
> We appreciate this suggestion. Following the algorithmic presentation style of Kim et al. (2024), we have added detailed pseudocode (Algorithm 2, Supplementary material) that clearly highlights the key differences.
>
>
>
>
> **Novelty relative to Kim et al.~(2024).**
>
> We emphasize that Nash is not a modification of Kim et al. but a generalization of empirical Bayes regression that introduces a new splitting variable enabling decoupled optimization. This change allows Nash to scale to
> $p=500,000+$ , whereas directly applying Kim et al. with a neural prior would require  p  neural updates per iteration, making it computationally infeasible.
>
>
>
>
> **Clarification of the GNN illustration (Fig.~1, bottom left).**
> This figure illustrates a simplified GNN architecture where we fix certain parameters (specifically s₂) to demonstrate the mathematical relationship between our framework and fused-lasso penalties (Equation 23). This is a pedagogical example showing how Nash can recover structured penalties as special cases. We have revised the caption and added text in Section 4.2 to clarify that this is an illustrative example, not a pre-trained model, and that the full experiments use learnable GNN parameters.
>
> **Additional experiment.**
>
> We have added a CNN comparison for MNIST denoising (Supplementary Figure 3, Section 3.4). The CNN was trained in a supervised manner on noisy-clean image pairs (see Supplementary Section 3.4, "CNN" paragraph, for architecture details). While the CNN achieves approximately 50% better RMSE than other methods, this comparison highlights a critical advantage of Nash: **the CNN requires access to clean target images during training, whereas Nash and other methods operate purely on noisy observations.** In practical denoising applications, clean ground truth is often unavailable, making Nash's unsupervised approach more broadly applicable. The CNN results thus establish an approximate performance ceiling for supervised methods while demonstrating that Nash achieves competitive performance without this privileged information.
>
>
>
> We hope these clarifications address the reviewer’s concerns and that you may consider increasing your evaluation score.

---

### Official Review · Reviewer_4eXJ · 2025-11-02

**Soundness:** 3
**Presentation:** 2
**Contribution:** 2
**Rating:** 4
**Confidence:** 4

**Summary:**

The paper proposes a new Bayesian method for linear regression.
The prior distribution of the coefficients has product form. Each coefficient \beta_j has prior that is the convolution of a Gaussian with variance \sigma_0^2, and a distribution that depends on some side information d_j and is parametrized by common parameters \theta.
The paper describes a variational method for estimating the parameters of the prior, and performing inference on the linear model coefficients:
- The posterior is approximated by a product posterior.
- The corresponding ELBO is  maximized via coordinate ascent.
The paper illustrates the application of their approach to various regression problems, and how the new prior introduced in this paper can
model a variety of complex structures (in particular group- and graph- based penalies) and demonstrates effectiveness on a denoising problem.

**Strengths:**

1. Regression with structural information about the coefficients is an central problem in high dimensional statistics with countless applications. Any progress on this problem is welcome.
2. The proposed framework is very general.
3. Simulations demonstrate some promising results.

**Weaknesses:**

1. The prior construction seems a straightforward extension of Wang & Stephens (2021) and Kim et al. (2024). The main innovation is the introduction of the "side information" d_j. While this is helpful, especially for graph-based tasks considered here, it is a very natural idea.
2. The variational inference algorithm is an application of standard methodology.

**Questions:**

1. I think Section 4, 5 describing the application and empirical results the most important of the paper. I think they should be expanded, spelling out in each case what is the architecture, how the prior was constructed, what are the x's and y's, what are the d's and so on.

2. In contrast, Fig. 1 is fairly obvious/ not informative (and could be removed for reasons of space). Same consideration for the bottom panel of Fig 2

Minor:

1. Eq (8), you should write what expectation is over

2. Also the subscript Nash to the ELBO appears in random positions.

---

> ### Author Response · Authors · 2025-11-19
>
> We thank the reviewer for the time and constructive feedback.
>
>
> **Novelty and relation to prior work.**
> We respectfully disagree that the extension is straightforward.
> While incorporating side information $d_j$ may appear natural in hindsight, the key challenges we address are:
>
> 1. Computational tractability: Direct application of prior work (Wang & Stephens 2021, Kim et al. 2024)
> with neural networks is computationally prohibitive.
> Our split-VEB formulation reduces computational cost from O(pK) neural network evaluations per iteration to O(p + K),
> enabling practical application.
> 2. Methodological innovation: Unlike Kim et al. (2024), which uses a two-step procedure with fixed hyperparameters, our split formulation enables joint optimization of all parameters while maintaining computational efficiency.
> This is not merely an implementation detail but fundamentally changes what models are practically feasible.
> 3. Scope: We demonstrate that a single unified framework can recover group lasso, fused lasso,
> and IPF-lasso as special cases, which has not been previously shown.
> We agree that the variational inference algorithm uses standard techniques,
> but the challenge was adapting these techniques to work efficiently with expressive,
> learned priors—a non-trivial contribution.
>
> **Clarifications to Sections 4 and 5.**
>
>  We agree that detailed experimental specifications are essential. Given the 9-page limit, we have chosen  to expanded the supplementary material fully specify the prior architecture, side information. We hope that these additions  improve transparency and reproducibility.
>
> **Minor comments.**
> We corrected the ELBO notation and clarified the expectation in Eq. (8).
>
>
> We hope these clarifications address the reviewer’s concerns and that you may consider increasing your evaluation score.

---

### Official Review · Reviewer_976x · 2025-11-07

**Soundness:** 3
**Presentation:** 3
**Contribution:** 2
**Rating:** 6
**Confidence:** 2

**Summary:**

This paper introduces Nash (Neural Adaptive Shrinkage), a framework for high-dimensional sparse regression that leverages covariate-specific side information through neural networks to learn adaptive penalties. The authors develop a split variational empirical Bayes (VEB) algorithm that decouples prior learning from posterior inference. The paper establishes connections to mr.ash and demonstrates competitive performance on four real datasets.

**Strengths:**

S1. The split VEB approach addresses a real computational bottleneck of  the mr.ash variational formulatio. By decoupling the updates of problems, Nash requires only one neural network update per coordinate ascent iteration per updates. Theorem A.1 provide the lower-bound relationship to mr.ash.

S2. The authors successfully demonstrates how Nash can encompass various structured penalties (group lasso, fused lasso, IPF-lasso) within a single framework.

S3. The authors clearly explains the variational formulation, coordinate ascent updates, and connections to exiting  variational Empirical Bayes approach.

**Weaknesses:**

W1. The authors claim Nash is "the first work to propose the use of a neural network to incorporate covariate side information when learning the penalty function," but fail to demonstrate why neural networks are necessary. They can compare more classical baselines including Kernel-based methods (e.g., RBF kernels on side information).

W2. The author employed only 4 real datasets, no synthetic data demonstrating when/why NNs help. It would be helpful if they can provide scenarios with complex non-linear side information where NN superiority would be clear.

W3. The theoretical contribution is limited to Theorem A.1. Although it shows Nash ≥ lower bound of mr.ash, but I want to understand How tight is this bound in practice.

**Questions:**

See weaknesses.

---

> ### Author Response · Authors · 2025-11-19
>
> We thank the reviewer for the  time and constructive feedback.
>
> **Regarding Weakness 1:**
>
> We appreciate the reviewer’s point about comparing against kernel-based methods such as those using RBF kernels on side information. We would like to highlight that the proposed Nash framework is model-agnostic—it can accommodate any mapping from side information to penalties, including kernel-based approaches, as long as the expectations in equation (5) can be evaluated or approximated.
>
> **Regarding Weakness 2:**
>
> We would like to highlight our Nash-fused experiment on MNIST denoising, where the side information is spatial. This setting naturally captures complex nonlinear dependencies. The model employs a Graph Neural Network to leverage spatial structure and smooth estimates using the image graph, effectively demonstrating the benefit of nonlinear representations of side information.
>
> **Regarding Weakness 3:**
>
> We have amended the supplementary material section A.4 (comparison with mr.ash) to include examples comparing the ELBO values of Nash and mr.ash. These plots provide additional insight into the tightness of the lower bound relationship established in Theorem A.1., which appear to be rather lose, NB: We show that Nash is a lower bound for mr.ash (and not an upper bound as written by the reviewer which we suspect to be a typo).
>
> Thank you for these suggestions, which has improved the paper.
> We hope these clarifications address the reviewer’s concerns and that you may consider increasing your evaluation score.

---

### Note · Authors · 2026-01-26

I have read and agree with the venue's withdrawal policy on behalf of myself and my co-authors.

---

### Meta-Review · Area_Chair_6J4X · 2026-01-04

**Summary:**

Reviewers generally agreed that the problem setting is important and that the paper is technically sound, but there was significant disagreement about the level of novelty and strength of contribution for ICLR. Several reviewers viewed Nash as a natural extension of existing empirical Bayes and variational regression frameworks (e.g., mr.ash and related work), with the main new element being the use of neural networks to map side information to penalties. For these reviewers, this step felt incremental rather than fundamentally new, and the variational algorithm was seen as an application of standard techniques. Concerns were also raised about the limited experimental scope: only a small number of real datasets were used, there was little synthetic analysis to clarify when neural networks are truly necessary, and performance gains were sometimes modest or insufficiently contextualized (e.g., lack of significance analysis). Overall, while some reviewers found the method promising, others felt the paper did not yet make a strong enough case that it represents a clear conceptual or empirical advance over existing approaches.

**Reviewer Concerns:**

The rebuttal addressed several points effectively. The authors clarified the computational motivation for the split VEB formulation, explained more clearly how Nash generalizes existing penalty structures, added pseudocode and additional experimental details in the supplement, and responded thoughtfully to questions about interpretability and the role of neural networks. These responses improved clarity and helped position the work more carefully relative to prior literature. However, the main outstanding issues remain. In particular, skeptical reviewers were still unconvinced that neural networks are essential rather than a flexible but unnecessary choice, that the empirical gains are strong and general enough to justify the added complexity, or that the contribution rises beyond a well-engineered extension of existing empirical Bayes methods. The experimental evidence still feels limited in scope, and the theoretical contribution remains relatively modest.

**Reviewer Scores:**

If reviewers were able to update scores, I expect only minor shifts. The more positive reviewers (scores around 6) would likely remain at similar levels, perhaps with slightly higher confidence after the clarifications. Reviewers who were marginally negative (scores around 4) might stay roughly where they are, as their concerns about novelty and experimental strength are only partially addressed. I do not expect the more skeptical reviewers to move substantially upward, since their core reservations about the incremental nature of the contribution and the necessity of neural components remain. As a result, the overall score distribution would likely remain mixed, leading me to recommend rejection.

---

### Decision · Program_Chairs · 2026-01-26

Reject